# ER-phagy restrains inflammatory responses through its receptor UBAC2

Xing He [ID] [1], Haowei He[1], Zitong Hou[1], Zheyu Wang[1], Qinglin Shi[1], Tao Zhou[1], Yaoxing Wu[2], Yunfei Qin[3], Jun Wang[4], Zhe Cai [ID] [5], Jun Cui [ID] [1] & Shouheng Jin [ID] [1 ✉]

## Abstract

ER-phagy, a selective form of autophagic degradation of endoplasmic reticulum (ER) fragments, plays an essential role in governing ER homeostasis. Dysregulation of ER-phagy is associated with the unfolded protein response (UPR), which is a major clue for evoking inflammatory diseases. However, the molecular mechanism underpinning the connection between ER-phagy and disease remains poorly defined. Here, we identified ubiquitin-associated domain-containing protein 2 (UBAC2) as a receptor for ER-phagy, while at the same time being a negative regulator of inflammatory responses. UBAC2 harbors a canonical LC3-interacting region (LIR) in its cytoplasmic domain, which binds to autophagosomal GABARAP. Upon ER-stress or autophagy activation, microtubule affinity-regulating kinase 2 (MARK2) phosphorylates UBAC2 at serine (S) 223, promoting its dimerization. Dimerized UBAC2 interacts more strongly with GABARAP, thus facilitating selective degradation of the ER. Moreover, by affecting ER-phagy, UBAC2 restrains inflammatory responses and acute ulcerative colitis (UC) in mice. Our findings indicate that ER-phagy directed by a MARK2-UBAC2 axis may provide targets for the treatment of inflammatory disease.

**Keywords** ER-phagy; UBAC2; MARK2; Inflammatory Responses; Colitis
**Subject Categories** Autophagy & Cell Death; Organelles

## Introduction

The endoplasmic reticulum (ER) governs the synthesis and transport of protein and lipid, the balance of iron, and the communication of sub-cellular organelles in eukaryotic cells (Gubas and Dikic, 2022). The morphology of the ER is highly dynamic and undergoes constant remodeling, thus modulating the size and activity of ER to maintain the cellular homeostasis (Foronda et al, 2023; Wiseman et al, 2022). The protein-handling capacity of ER is compromised under diverse extracellular and intracellular conditions, causing a state of 'ER stress' to activate the signaling cascade named as unfolded protein response (UPR) (Di Conza et al, 2023). UPR either restores the homeostasis of the ER or induces the death of the cell when the damage is irreparable (Hetz et al, 2020b). In addition to UPR, selective autophagy has been demonstrated to keep the quality control of ER, which termed as ER-phagy or reticulophagy (Jiang et al, 2020).

ER-phagy is largely activated during a variety of stress, including accumulation of misfolded protein, perturbation of intracellular calcium content, and alteration in redox status (Gubas and Dikic, 2022). Serving as a main recovery mechanism, ER-phagy degrades protein aggregates within both the lumen and the surface of ER (Lipatova et al, 2020). ER-phagy employs receptors that connect domains of ER sheets or tubules to the autophagy machinery by binding to members of ATG8 protein family, which is comprised of LC3 and GABARAP proteins in mammals (Khaminets et al, 2015). During the process of ER-phagy, a number of ER surface proteins, such as ATL3, FAM134B, RTN3L, and TEX264, function as specific receptors through LC3-interacting regions (LIRs) to bring ER fragment to autophagy machinery (Chen et al, 2019; Chino et al, 2019; Forrester et al, 2019; Zielke et al, 2021). The ER is then sequestered within lysosomes for efficient degradation by lysosomal enzymes (Yamamoto et al, 2023). But beyond the few receptors have been reported, our knowledge of the signals that regulate the initiation and precise control of ER-phagy is relatively limited.

Emerging evidence shows that ER homeostasis has significant effects on maintaining optimal host immune responses. Multiple branches of the UPR provoke IL-6 expression relying on pattern recognition receptor (PRR) signaling in the context of immune and metabolic status. The branch of UPR through the cleavage of mRNA encoding X-box protein 1 (XBP1) to produce the spliced transcription factor XBP1s, can mediate the release of interferon-β (IFNβ) via IRF3 through TLR3 signaling activation (Hu et al, 2011). XBP1s overexpression increases the production of IFNβ and suppresses vesicular stomatitis virus (VSV) infection (Di Conza et al, 2023). Stimulator of interferon genes (STING) localizes in the ER and senses cyclic GMP–AMP (cGAMP) (Sun et al, 2013). Upon challenge with Gram-positive bacteria, STING triggers the UPR through the protein kinase RNA-like ER kinase (PERK) activation

[1]Guangdong Province Key Laboratory of Pharmaceutical Functional Genes, MOE Key Laboratory of Gene Function and Regulation, State Key Laboratory of Biocontrol, School of Life Sciences, Sun Yat-sen University, Guangzhou, China. [2]Institute of Precision Medicine, Department of Critical Care Medicine, the First Affiliated Hospital, Sun Yat-sen University, Guangzhou, China. [3]Biotherapy Center, The Third Affiliated Hospital, Sun Yat-sen University, Guangzhou, China. [4]Precision Research Center for Refractory Diseases, Institute for Clinical Research, Shanghai General Hospital, Shanghai Jiao Tong University School of Medicine, Shanghai, China. [5]Guangzhou Institute of Pediatrics, Guangzhou Women and Children's Medical Center, Guangzhou Medical University, Guangzhou, China. ✉E-mail: jinshh3@mail.sysu.edu.cn

in macrophages. The induction of ER-phagy eliminates stressed ER membranes and contributes to produce protective type I IFN. The deficiency of ER-phagy results in unresolved ER stress and the death of bacteria-infected phagocytes (Moretti et al, 2017). The branch of UPR via inositol-requiring enzyme 1α (IRE1α) signaling activates TRAF2 within the ER membrane to trigger inflammatory responses through the NF-κB pathway (Keestra-Gounder et al, 2016). Moreover, NOD1 and NOD2 are crucial mediators of inflammation derived from ER stress. Infection with *Citrobacter rodentium* leads to the activation of IRE1α-NOD1 and/or IRE1α-NOD2, which subsequently promotes inflammation and bacterial clearance (Sweet et al, 2022). The growing links between ER-phagy and immunity have revealed that ER-phagy can provide potential therapy targets, however, the detailed mechanism underpinning the crosstalk between ER-phagy and immune response remains elusive.

The gene encoding ubiquitin-associated domain-containing protein 2 (UBAC2) is highly conserved in different species (Han et al, 2019; Zhou et al, 2018). *UBAC2* is a risk allele of Behçet's disease and the enhanced UBAC2 expression prompts the progress of Behçet's disease (Fei et al, 2009). UBAC2 has also been reported to be highly correlated with the development of malignant tumors (including skin and bladder cancer) and inflammatory bowel diseases (Gu et al, 2020; Misselwitz et al, 2021; Nan et al, 2011). UBAC2 is an ER-resident protein which contains 3 transmembrane domains, whether it participates in the regulation of ER homeostasis is still uncertain. Here, using a rapid and robust proximity labeling system termed as *Pyrococcus horikoshii* biotin protein ligase (*Ph*BPL)-assisted biotin identification (PhastID), we captured endogenous proteins on the endoplasmic reticulum membrane (ERM) and identified UBAC2 serves as a novel ER-phagy receptor. UBAC2 undergoes autophagic degradation under starvation- or ER stress-induced conditions. UBAC2 harbors a highly conserved LIR motif and interacts with GABARAP. Upon ER stress, microtubule affinity-regulating kinase 2 (MARK2) phosphorylates UBAC2 at serine (S) 223 to promote the dimerization of UBAC2. UBAC2 dimerization enhances the association between UBAC2 and GABARAP, thus accelerating the progression of ER-phagy. Moreover, *UBAC2* deficiency results in the inflammatory responses through disrupting the ER homeostasis. UBAC2 variants from inflammatory diseases or point mutation in the LIR motif of UBAC2 decrease ER-phagy flux and increase sterile inflammation associated with ER stress in vivo, which makes mice more susceptible to dextran sulfate sodium (DSS)-induced ulcerative colitis (UC). Our findings demonstrate that UBAC2 functions as an ER-phagy receptor to maintain optimal immunity by balancing ER-phagy and inflammatory responses.

# Results

## UBAC2 undergoes autophagic degradation

To gain insight into the maintenance of ER homeostasis, we designed an unbiased proteomic approach to identify proteins that are recruited to the ER membrane (ERM). We performed an improved BioID system termed PhastID that labels ER within short-term of biotin addition through targeting biotin ligase to the ERM by fusing it to the ERM anchor derived from cytochrome P450 (Branon et al, 2018; Feng et al, 2024). The proteins in the ER

membrane (ERM)-facing cytosol can be identified by streptavidin (Fig. 1A). We purified biotinylated ERM-associated proteins from HeLa cells with or without the treatment of thapsigargin (TG, an ER stressor) for mass spectrometry (MS) analysis. The protein abundance of a previous reported ER stress inducible protein (Smith et al, 2018), cell cycle progression protein 1 (CCPG1) was upregulated in our results, confirming the utility of our approach. In addition, we surprisingly identified that the protein level of UBAC2, an ER resident protein which is highly related with several diseases, was attenuated upon TG treatment (Fig. 1B). The transcription of *UBAC2* in the same cells remains unchanged after treatment with TG (Fig. 1C). We further studied the degradation system that dominantly controlled UBAC2 turnover under TG treatment, and observed the UBAC2 degradation mediated by TG was blocked by the autolysosome inhibitor bafilomycin A1 (Baf A1), but not the proteasome inhibitor MG132 or carfilzomib (Carf) (Fig. 1D,E). Our results suggested that TG prompted the degradation of UBAC2 through autophagy process. We next found that the protein abundances, but not the mRNA abundances of UBAC2 were gradually reduced with TG treatment in a manner that is dose-dependent (Fig. 1F,G; Appendix Fig. S1A–C). Moreover, the degradation of UBAC2 induced by starvation-mediated autophagy activation under Earle's balanced salt solution (EBSS) culture condition was almost totally abrogated in the presence of Baf A1 in HeLa cells, THP-1-derived macrophages (THP-1 Mφs) and HT-29 cells (Fig. 1H; Appendix Fig. S1D–F).

The EBSS-triggered UBAC2 degradation was disappeared in *BECN1* knockout (KO) cells, in which the canonical autophagy process is dramatically impaired (Fig. 1I; Appendix Fig. S1G). Nascent autophagosome is generated from the membranes with phosphatidylinositol 3-phosphate (PI3P) to recruit WIPI2, thereby recruiting ATG16L1 for LC3-conjugation (Puri et al, 2018). To further investigate how UBAC2 transmits through autophagic process, we observed that EBSS-induced starvation increased the co-localization between UBAC2 and WIPI2[+] as well as ATG16L1[+] puncta (Fig. 1J–M). In addition, autophagy activation promoted the distribution of UBAC2 into LAMP1 (lysosomal-associated membrane protein 1)[+] compartments (Fig. 1N,O). Coimmunoprecipitation (Co-IP) and immunoblot analysis showed that the interaction between UBAC2 and WIPI2, ATG16L1, or LAMP1 was increased under the activation of starvation-induced autophagy (Appendix Fig. S1H). Taken together, these data demonstrate that UBAC2 undergoes degradation under ER stress or starvation condition in a canonical autophagy-dependent manner.

## UBAC2 is an LIR motif-containing interactor of GABARAP

Considering the importance of WIPI2 and ATG16L1 for the lipidation of ATG8-family proteins, and the elongation of nascent autophagosomes, we moved on to study whether UBAC2 interacts with ATG8-family proteins. Co-IP and immunoblot analysis indicated that UBAC2 interacted with LC3 (MAP1LC3A/B/C), GABARAP, and GABARAPL2, while the association between UBAC2 and GABARAP was the strongest (Figs. 2A and EV1A–C). The UBAC2-GABARAP association was significantly increased upon both TG and EBSS-treated conditions (Fig. EV1D,E). We treated HeLa cells with TG or EBSS medium and collected cell lysates at varying time points, and found that the endogenous

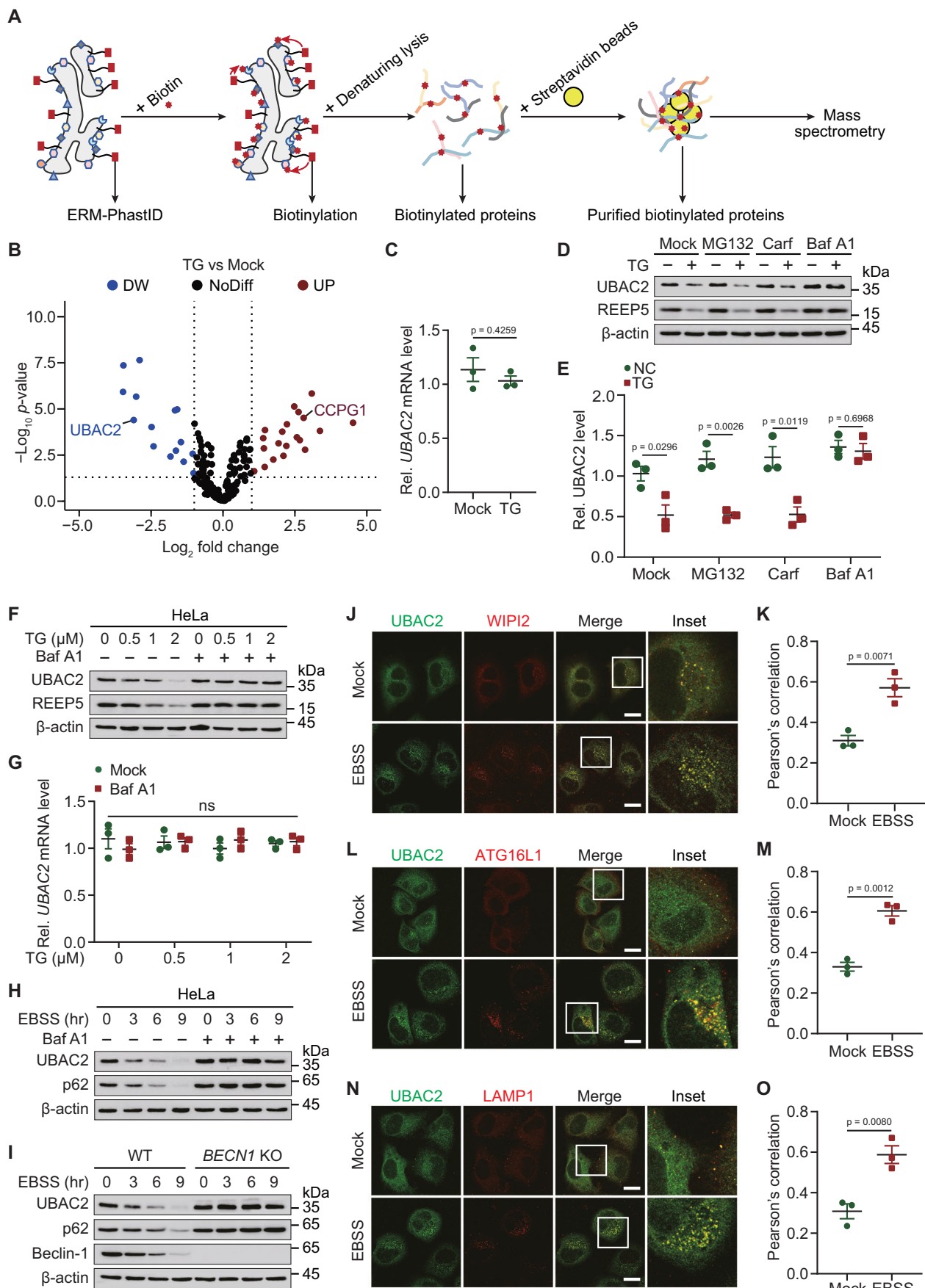

**Figure 1. UBAC2 can be degraded through autophagy process.**

(A) Schematic illustration of the ERM-PhastID screen. (B) HeLa cells expressing ERM-PhastID were treated with or without thapsigargin (TG) (1 μM) for 2 h, and cultured within biotin (50 μM) for 1 h. The biotinylated proteins were collected by streptavidin-coated beads, and identified using mass spectrometry (MS). This experiment was performed once with two replicates per condition. (C) The mRNA levels of *UBAC2* in HeLa cells expressing ERM-PhastID treated with or without TG (1 μM) for 3 h. (D) HeLa cells cultured in the absence or presence of TG (1 μM) for 3 h were treated with MG132 (10 μM), carfilzomib (10 μM), or bafilomycin A1 (0.2 μM) for 6 h. The cell extracts were analyzed by immunoblotting. (E) The quantification of UBAC2 protein abundance in the similar sample as (D) from three independent experiments performed in duplicate. (F) HeLa cells treated with TG with indicated dosages for 3 h in the absence or presence of Baf A1 (0.2 μM), and the proteins were collected for immunoblot analysis. (G) The mRNA levels of *UBAC2* in the similar sample as (F). (H) HeLa cells were cultured in EBSS for indicated time points with or without Baf A1 (0.2 μM) treatment, and the lysates were detected by immunoblot. (I) Lysates of wild-type (WT) and *BECN1* KO 293 T cells cultured in EBSS for indicated time points were subjected to immunoblot analysis. (J) HeLa cells cultured in the absence or presence of EBSS for 3 h in the presence of Baf A1 (0.2 μM), followed by labeling of UBAC2 (green) and WIPI2 (red) with specific antibodies. Scale bar, 20 μm. (K) Quantitative analysis of the similar samples as (J) from three biologically independent experiments (20 cells scored per condition per experiment). (L) HeLa cells cultured in the absence or presence of EBSS for 3 h in the presence of Baf A1 (0.2 μM), followed by labeling of UBAC2 (green) and ATG16L1 (red) with specific antibodies. Scale bar, 20 μm. (M) Quantitative analysis of the similar samples as (L) from three biologically independent experiments (20 cells scored per condition per experiment). (N) HeLa cells cultured in the absence or presence of EBSS for 3 h in the presence of Baf A1 (0.2 μM), followed by labeling of UBAC2 (green) and LAMP1 (red) with specific antibodies. Scale bar, 20 μm. (O) Quantitative analysis of the similar samples as (N) from three biologically independent experiments (20 cells scored per condition per experiment). Data information: For (D, F, H–J, L, N), one representative experiment out of three was shown. In (C, E, G, K, M, O), data are presented as the mean ± SEM of three independent biological experiments. The statistical significance of the difference was analyzed by unpaired two-tailed Student's *t* test, and the *P* values were shown. Source data are available online for this figure.

association between UBAC2 and GABARAP increased considerably upon TG or EBSS treatment (Figs. 2B,C and EV1F,G). Confocal analysis further indicated that UBAC2 colocalized with GABARAP during autophagy activation (Fig. 2D,E). To further unveil the mechanism of GABARAP in modulating the autophagic degradation of UBAC2, we found that the cytosolic domain of UBAC2 exhibits a LIR motif (WNRL) (Figs. 2F and EV1H). Since the LIR motif is important for the association between autophagy adaptors and ATG8-family proteins, we generated a UBAC2 LIR mutant (LIRm) harboring W275A and L278A, finding that UBAC2 LIRm failed to interact with GABARAP (Figs. 2G and EV1I). Confocal imaging revealed that the co-localization of GABARAP and UBAC2 mutant was also abrogated (Fig. 2H,I). We then performed in vitro His-pulldown assays and observed that purified wild-type (WT) UBAC2, but not UBAC2 LIRm could directly bind to GABARAP (Fig. 2J). In line with the abolished autophagic degradation of UBAC2 LIRm (Figs. 2K and EV1J), the co-localization between UBAC2 LIRm and LAMP1 was dramatically decreased (Fig. 2L,M). It has been reported that the GABARAP Y25A or K46A mutant disrupted the interaction between GABARAP and substrates (Chen et al, 2019). Consistently, we found that the UBAC2-GABARAP association was largely decreased when the Y25 or K46 was mutated both in vitro and in vivo (Figs. 2N and EV1K). Altogether, these data indicate that UBAC2 is degraded through autophagy as a LIR motif-containing protein.

## UBAC2 serves as an ER-phagy receptor

As UBAC2 is an ER transmembrane protein and can be delivered to autolysosomes through directly binding with GABARAP, we speculated that UBAC2 works as an ER-phagy receptor. We adopted a doxycycline (Dox)-inducible ER-phagy reporter, which contains an N-terminal ER signal sequence (SS) followed by tandem monomeric RFP and GFP sequences and the ER retention sequence KDEL as previously described (Chino et al, 2019) (Fig. 3A). Knocking out *UBAC2* using CRISPR-Cas9 method significantly reduced the production of RFP fragment under starvation-induced autophagy condition or TG treatment in HeLa cells (Figs. 3B,C and EV2A,B). We also generated *UBAC2* KO THP-1 and HT-29 cells, observing that

*UBAC2* depletion attenuated the ER-phagy flux under starvation-induced autophagy or TG-induced ER-stress conditions (Fig. EV2C–F). We also monitored the ER-phagy activity by fluorescence microscopy and observed that the number of red puncta was smaller in *UBAC2* KO cells than that in WT cells upon starvation or ER-stress treatment (Figs. 3D,E and EV2G,H). In addition, the ER content was increased by *UBAC2* depletion (Fig. 3F). We found that only WT UBAC2, but not UBAC2 LIRm promoted ER-phagy flux and the reduction of ER content (Figs. 3G–I and EV2I,J), suggesting that UBAC2 affects the ER-phagy activity through interacting with GABARAP.

For better observing the ER compartment, we employed transmission electron microscopy analysis and found that *UBAC2* KO led to a striking ER expansion especially in the cell periphery, while this phenomenon could be recovered by WT UBAC2, but not UBAC2 LIRm (Fig. 3J). To clarify the relationship between UBAC2 and other known ER-phagy receptors, we performed Co-IP and immunoblot analysis and observed that UBAC2 only weakly interacted with FAM134B. However, the interaction between UBAC2 and FAM134B was not affected by starvation-induced autophagy activation (Fig. 3K). To further study whether FAM134B is essential for the ER turnover mediated by UBAC2, we depleted the expression of *FAM134B* and found that UBAC2-promoted ER-phagy was not affected in *FAM134B*-deficient cells (Fig. 3L). Our findings suggest that the role of UBAC2 in ER-phagy is not dependent on FAM134B, ATL3, SEC62, RTN3, CCPG1, and TEX264. Altogether, these data indicate that UBAC2 functions as a newly identified ER-phagy receptor.

## Phosphorylation is essential for the role of UBAC2 in ER-phagy

To figure out how UBAC2 selectively targets ER for degradation, we adopted MS and identified a phosphorylation site of UBAC2 at serine (S) 223 (Fig. 4A). To test the hypothesis that phosphorylation is selective signal for the regulation of UBAC2-mediated ER-phagy, we immunoprecipitated all phosphorylated proteins using phosphoserine/threonine/tyrosine polyclonal antibody from cellular lysates, and detected high apparent band of UBAC2 under both TG and EBSS treatment (Figs. 4B and EV3A). We then generated

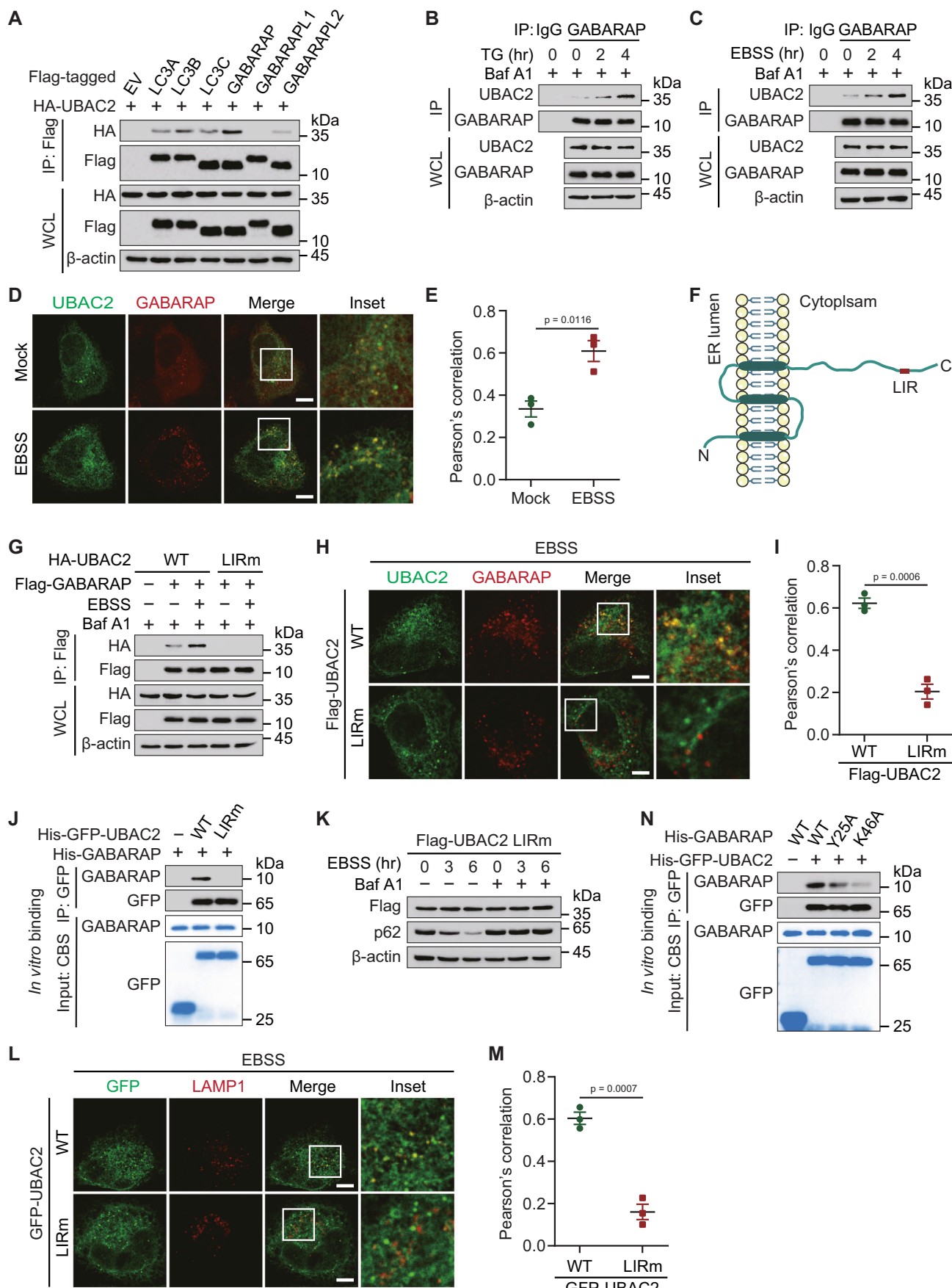

**Figure 2. UBAC2 interacts with GABARAP through its LIR.**

(A) HEK293T cells were transfected with plasmids encoding HA-UBAC2 and Flag-tagged ATG8 family members (Flag-LC3A, Flag-LC3B, Flag-LC3C, Flag-GABARAP, Flag-GABARAPL1, and Flag-GABARAPL2), followed by immunoprecipitation (IP) with anti-Flag beads and immunoblot analysis with anti-HA. Throughout was the immunoblot analysis of whole-cell lysates (WCL) without immunoprecipitation. (B, C) Extracts of HeLa cells incubated with TG (1 μM) (B) or EBSS (C) for various times were subjected to immunoprecipitation with anti-GABARAP and immunoblot analysis with indicated antibodies. (D) HeLa cells cultured in the absence or presence of EBSS for 3 h in the presence of Baf A1 (0.2 μM), followed by labeling of UBAC2 (green) and GABARAP (red) with specific antibodies. Scale bar, 20 μm. A representative experiment out of three is shown. (E) Quantitative analysis of the similar samples as (D) from three biologically independent experiments (20 cells scored per condition per experiment). (F) Structure of human UBAC2 depicting the identified LC3-interacting region (LIR). (G) Immunoprecipitation and immunoblot analysis of 293 T cells transfected with vectors encoding Flag-GABARAP and wild-type (WT) HA-UBAC2 or its LIR mutant (LIRm). (H) HeLa cells transfected with Flag-UBAC2 or its LIRm form in the presence of EBSS for 3 h with the existence of Baf A1 (0.2 μM), followed by labeling of Flag (green) and GABARAP (red) with specific antibodies. Scale bar, 20 μm. (I) Quantitative analysis of the similar samples as (H) from three biologically independent experiments (20 cells scored per condition per experiment). (J) His-GFP-UBAC2 and His-GABARAP were expressed and purified from *E. coli*, respectively. The purity of protein was analyzed by SDS-PAGE and visualized by Coomassie Blue staining (CBS). The His-GFP-UBAC2 was incubated with His-GABARAP in reaction buffer in vitro. After pull-down with GFP-beads, the bound material was analyzed by immunoblotting. (K) 293T cells were transfected with plasmids encoding Flag-UBAC2 LIRm, then were cultured in EBSS for indicated time points with or without Baf A1 (0.2 μM). The protein expression levels of UBAC2 were detected by immunoblot. (L) HeLa cells transfected with GFP-UBAC2 or its LIRm form in the presence of EBSS for 3 h with the existence of Baf A1 (0.2 μM), followed by labeling of LAMP1 (red) with specific antibodies. Scale bar, 20 μm. (M) Quantitative analysis of the similar samples as (L). (N) His-GFP-UBAC2 and His-GABARAP as well as its indicated mutants were expressed and purified from *E. coli*, respectively. GFP-UBAC2 was incubated with His-GABARAP in reaction buffer in vitro. After pull-down with GFP-beads, the bound material was analyzed by immunoblotting. Data information: For (A–D, G, H, J–L, N), one representative experiment out of three was shown. In (E, I, M), data are presented as the mean ± SEM of three independent biological experiments. The statistical significance of the difference was analyzed by unpaired two-tailed Student's *t* test, and the P values were shown. Source data are available online for this figure.

phosphoresistant form of UBAC2 by substituting S residue with alanine (A), observing that S223A prevented the phosphorylation of UBAC2 induced by TG or EBSS treatment (Figs. 4C and EV3B). Since many autophagy receptors undergo dimerization or oligomerization during autophagy for degradation (Jiang et al, 2020; Pan et al, 2016), we found that TG or EBSS treatment promote the self-interaction of UBAC2 (Figs. 4D and EV3C). We further investigated the assembly of UBAC2 oligomerization through chemical cross-linking using disuccinimidyl suberate (DSS) and consequent western blot showed that TG treatment induced the dimerization of UBAC2 in a dose-dependent manner (Fig. 4E). In addition, the formation of UBAC2 dimer was increased by TG or EBSS treatment over time (Figs. 4F and EV3D). Fluorescence microscopy assay also indicated that the UBAC2 puncta was accumulated upon Baf A1 or TG treatment (Fig. 4G–J).

We then constructed the phosphomimetic form of UBAC2 by substituting S residue with aspartate (D), and found that the S223D mutant form of UBAC2 had a stronger ability to form dimers, and the enhanced dimerization of UBAC2 caused by TG or EBSS treatment was almost abrogated when S223 was mutated to alanine or aspartate (Figs. 4K and EV3E). Consistently, UBAC2 S223A mutant displayed a smaller number of puncta, while UBAC2 S223D mutant had larger number of puncta (Fig. 4L,M). The cleaved RFP fragment was decreased by S223A mutant form of UBAC2, whereas increased by UBAC2 S223D mutant (Figs. 4N and EV3F). The association between UBAC2 S223A mutant and GABARAP was decreased while the interaction between UBAC2 S223D mutant and GABARAP was increased (Fig. EV3G). Furthermore, the decreased ER expansion in the cell periphery could not be recovered by overexpression of UBAC2 S223A mutant, while the S223D mutant form of UBAC2 enhanced the ER expansion (Fig. 4O). Collectively, all these data suggest that the phosphorylation of UBAC2 at S223 is crucial for the UBAC2-mediated ER-phagy process.

## MARK2 phosphorylates UBAC2 to enhance ER-phagy

To uncover the detailed mechanism of how UBAC2 undergoes phosphorylation, we employed MS using UBAC2 as a bait to search

the protein kinase of UBAC2. MS spectra corresponding to a specific peptide of microtubule affinity-regulating kinase 2 (MARK2) was present in anti-UBAC2 precipitates (Fig. 5A). We mainly focused on MARK2 due to its important role in protein misfolding stress (Lu et al, 2021), which is highly integrated with ER stress. We validated the MS results using coimmunoprecipitation assay, observing that both TG and EBSS treatment increased the interaction between UBAC2 and MARK2 (Figs. 5B and EV4A–C). We observed that purified UBAC2 from *E. coli* could interact with purified MARK2 from 293T cells (Fig. 5C) We then applied specific siRNAs targeting MARK2 (Fig. EV4D), observing that *MARK2* depletion abrogated the enhanced phosphorylation of UBAC2 upon TG or EBSS treatment (Figs. 5D and EV4E). The incubation of UBAC2 with MARK2 increased the phosphorylation of WT UBAC2, but not the S223A mutated form of UBAC2 in vitro (Fig. 5E). The dimerization of UBAC2 induced by TG or EBSS treatment was disappeared in *MARK2*-deficient cells (Figs. 5F and EV4F–H). The UBAC2 S223D mutant has a stronger ability to undergo dimerization, while the dimerization of S223A UBAC2 was largely diminished. Furthermore, the decreased UBAC2 dimerization induced by *MARK2* deficiency was abrogated when the S223 was mutated (Figs. 5G and EV4I). The phosphomimetic form of UBAC2 tends to form puncta (Fig. 5H,I). The interaction between GABARAP and UBAC2 was decreased by *MARK2* depletion. When the S223 was mutated, the reduced GABARAP-UBAC2 association induced by *MARK2* deficiency was almost abrogated (Fig. 5J). We then examine the ER-phagy with ss-RFP-GFP-KDEL reporter, finding that the enhanced production of RFP fragment by UBAC2 upon EBSS treatment was abrogated in *MARK2*-deficient cells (Figs. 5K and EV4J). Moreover, the decreased cleavage of RFP fragment by MARK2 depletion was abolished when the S223 was mutated in UBAC2 (Fig. EV4K,L). Immunoblotting study further revealed that UBAC2 S223D mutant, but not S223A mutated form of UBAC2 had a stronger ability to decrease ER content (Fig. 5L). We detected the activation of MARK2 signaling and found that PEP005, a PKCδ activator treatment could promote the activation of the PKCδ-MARK2 signaling axis and ER-phagy flux. However, the PEP005 induced

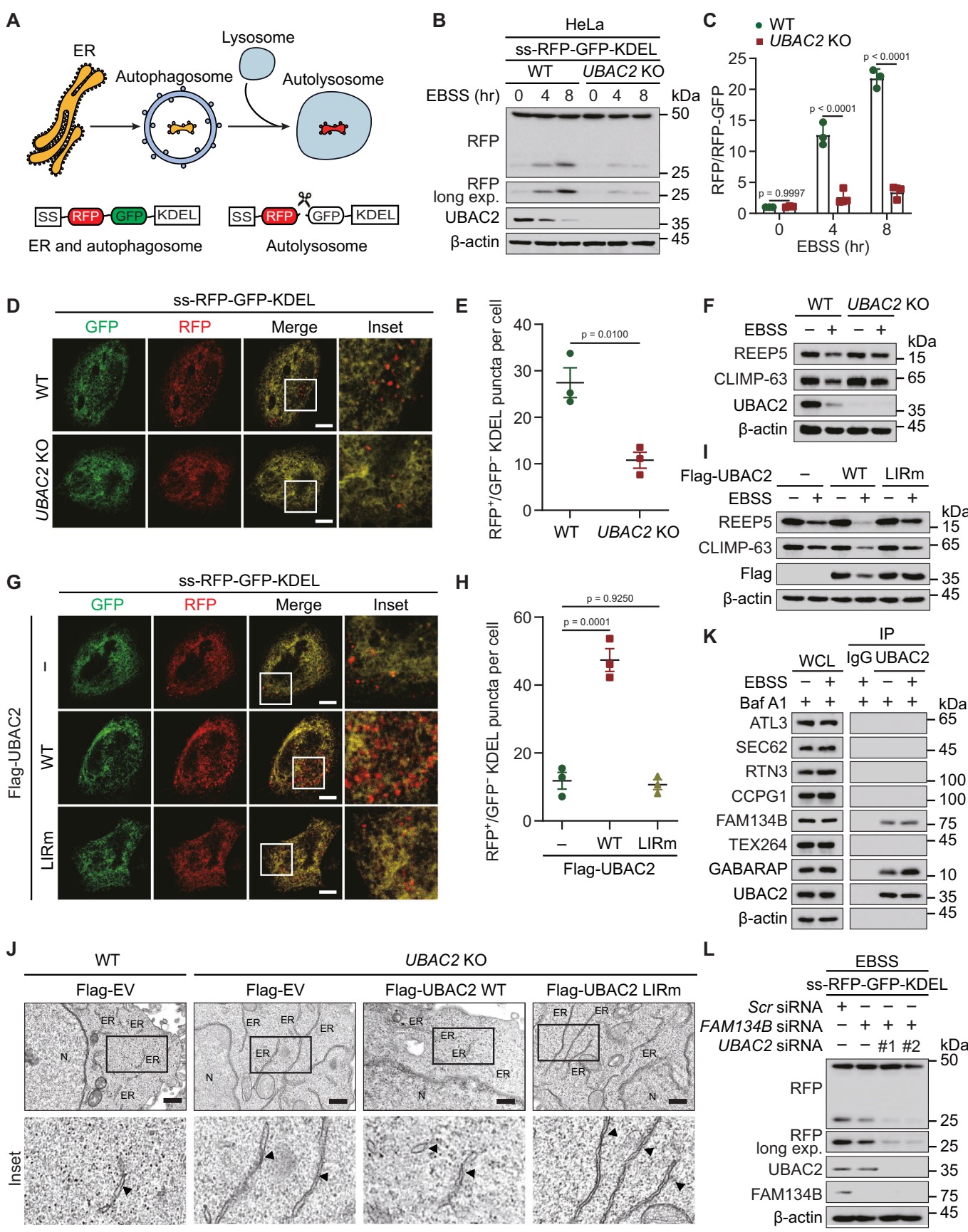

**Figure 3.  UBAC2 is an ER-phagy receptor.**

(A) Schematic representation of the ER-phagy reporter ss-RFP-GFP-KDEL. ss-RFP-GFP-KDEL is cleaved by lysosomal enzymes to yield the RFP fragment. The GFP signal is quenched in lysosomes. SS, signal sequence. (B) WT and *UBAC2* KO HeLa cells stably expressing the ER-phagy reporter were cultured in the presence of doxycycline (Dox) (200 ng/mL) for 24 h to induce the reporter. After cultured with EBSS for indicated time points, the protein was harvested for immunoblot analysis. (C) The band intensities of RFP and RFP-GFP were quantified and the ratio of RFP/RFP-GFP is shown. (D) WT or *UBAC2* KO HeLa cells stably expressing the ER-phagy reporter were treated with Dox (200 ng/mL) for 24 h. After cultured in EBSS for 3 h, the cells were processed and observed by fluorescence microscopy. Scale bar, 20 μm. (E) Quantitative analysis of the similar samples as (D) from three biologically independent experiments (20 cells scored per condition per experiment). (F) Lysates from WT or *UBAC2* KO HeLa cells cultured in EBSS were analyzed by immunoblot. (G) HeLa cells stably expressing the ER-phagy reporter were transfected with plasmids encoding WT or LIRm UBAC2 and treated with Dox (200 ng/mL) for 24 h. After cultured in EBSS for 3 h, the cells were observed by fluorescence microscopy. Scale bar, 20 μm. (H) Quantitative analysis of the similar samples as (G) from three biologically independent experiments (20 cells scored per condition per experiment). (I) HeLa cells transfected with plasmids expressing WT UBAC2 or UBAC2 LIRm were cultured in normal medium or EBSS for 3 h. The lysates were harvested for immunoblotting analysis. (J) Transmission electron microscopy of WT and *UBAC2* KO HeLa cells transfected with vector of Flag-UBAC2 or its LIRm and treated with EBSS for 3 h. The ER was indicated by black arrowheads. Scale bar, 200 nm. (K) Extracts of HeLa cells incubated with EBSS for 3 h in the presence of Baf A1 (0.2 μM) were subjected to immunoprecipitation with anti-UBAC2 and immunoblot analysis with indicated antibodies. (L) HeLa cells stably expressing the ER-phagy reporter were transfected with scramble, *FAM134B-*, or *UBAC2*-specific siRNA cultured in the presence of Dox (200 ng/mL) for 24 h to induce the reporter. After treated with EBSS for 3 h. The lysates were collected for immunoblot analysis. Data information: For (B, D, F, G, I–L), one representative experiment out of three was shown. In (C, E, H), data are presented as the mean ± SEM of three independent biological experiments. The statistical significance of the difference was analyzed by unpaired two-tailed Student's *t* test, and the *P* values were shown. Source data are available online for this figure.

ER-phagy flux was significantly abolished in UBAC2-deficient cells (Fig. EV4M). Taken together, these data indicate MARK2 catalyzes the phosphorylation of UBAC2 at S223, which is critical for promoting the dimerization of UBAC2 to further ER-phagy.

## UBAC2 prevents ER stress-associated inflammatory responses

ER stress is a cellular response initiated by an accumulation of misfolded proteins in the ER. ER-phagy suppresses ER stress via targeting the ER for selective autophagic degradation (Huang et al, 2019). ER stress is activated through three pivotal branches of UPR: IRE1α, PERK, and ATF6. To study whether UBAC2-mediated ER-phagy modulates the ER homeostasis, we generated UBAC2- and UBAC2 LIRm-inducible THP-1 cell lines to investigate the role of UBAC2 in UPR (Fig. 6A,B). We detected the transcription levels of spliced *XBP1*, which occurs downstream of the IRE1α pathway, alongside with *CHOP*, and *BiP*, additionally ER stress induced genes (Koksal et al, 2021). qPCR analysis revealed that over-expression of WT UBAC2, but not UBAC2 LIRm could inhibit the expression of ER-stress induced transcripts under tunicamycin (TM) treatment (Fig. 6C). Consistently, the transcription of ER-stress induced genes was increased in *UBAC2* KO cells when they were challenged with TM (Fig. 6D). To further confirm the above results, we employed the reversible inhibitor of the sarcoendoplasmic reticulum pump Ca²⁺ ATPase (SERCA) pump, cyclopiazonic acid (CPA), to induce ER stress. Our data showed that only WT UBAC2 restrained the transcription of ER-stress induced genes under CPA or TG treatment, and *UBAC2* KO enhanced the expression of ER-stress induced transcripts (Appendix Fig. S2A,B).

The transcription of inflammatory cytokines induced by TM, TG or CPA treatment could be suppressed by overexpression of WT UBAC2, but not UBAC2 LIRm (Fig. 6E; Appendix Fig. S2C). *UBAC2* KO increased the mRNA levels of inflammatory cytokines induced by TM, TG or CPA treatment (Fig. 6F; Appendix Fig. S2D). In line with the transcription of inflammatory cytokines, enzyme linked immunosorbent assay (ELISA) revealed that the release of IL6, TNFα, and IL-1β was suppressed by the overexpression of WT UBAC2. Moreover, *UBAC2* KO increased the production of IL6, TNFα, and IL-1β (Fig. 6G,H; Appendix Fig. S2E,F). We performed

western blotting analysis and found that *UBAC2* KO increased the protein abundances of ER stress markers upon TG or TM treatment. We also measured the activation of NF-κB signaling pathway, observing that *UBAC2* depletion enhanced the phosphorylation of p65 induced by TG or TM. Moreover, we found that *UBAC2* KO increased the protein levels of IFIT1 and MX1 (both are encoded by ISGs). In addition, only WT UBAC2, but not LIR mutated UBAC2 reduced the protein abundances of ER stress markers, the phosphorylation of p65, alongside with the expression levels of ISG-encoded products (Appendix Fig. S2G). We then knocked down the expression of *MARK2* in UBAC2-inducible cell line, observing that the suppression of ER-stress induced transcripts by UBAC2 was abrogated by *MARK2* deficiency (Fig. 6I; Appendix Fig. S2H). Consistently, the inhibitory transcription of inflammatory genes by UBAC2 was also abolished in *MARK2*-deficienct cells (Fig. 6J; Appendix Fig. S2H). Altogether, these data indicate that UBAC2 serves as ER-phagy receptor to suppress UPR and ER stress-associated inflammatory responses.

## UBAC2 attenuates DSS-induced colitis in mice through ER-phagy

Dysregulation of ER-phagy leads to inflammatory responses and is associated with immune diseases (Chaudhary et al, 2022). Recombinant adeno-associated virus (AAV) vector-mediated gene delivery provides a new approach for gene therapy. We selected AAV serotype 9, as it has been previously reported with relatively high efficiency in gut transduction (Fang et al, 2019), and transfected 8-week-old C57BL/6 mice with increasing dosages of AAV-UBAC2-GFP through tail vein injection. Immunoblotting was used to detect GFP expression in intestines to show the intestinal transduction efficiency in vivo (Fig. EV5A). We constructed acute UC model to determine if UBAC2 provides protective role in mice, observing that UBAC2 alleviated the body weight loss of DSS mice in a dose-dependent manner (Fig. EV5B). Furthermore, UBAC2-overexpressed DSS mice significantly slowed down the shortening of colon lengths, and displayed much better consistency and morphology of the colons without inflammatory infiltration (Fig. EV5C–F). We harvested the intestines for qPCR analysis and found that UBAC2 overexpression suppressed the

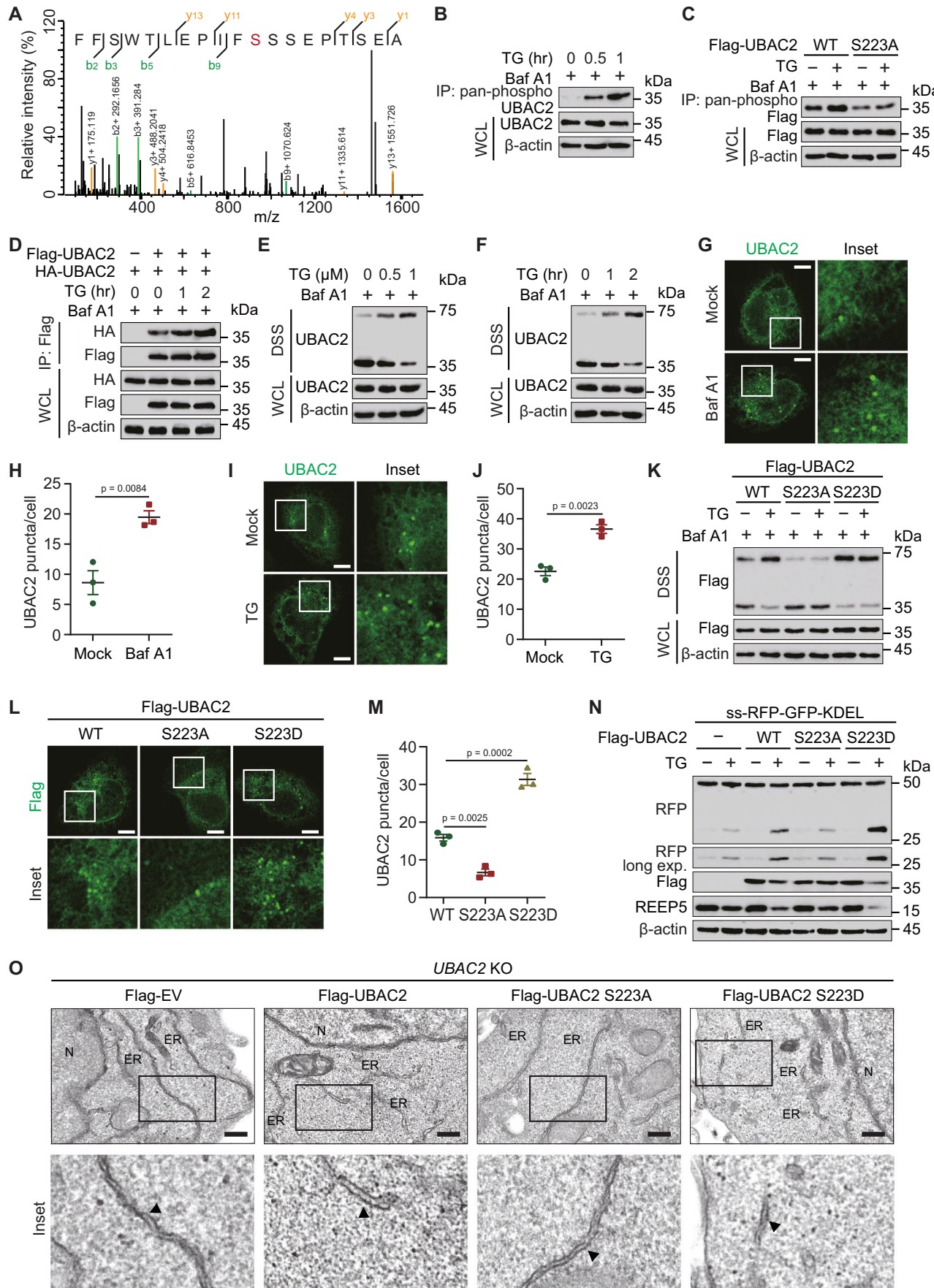

Figure 4.   UBAC2 phosphorylation at S223 is essential for its role in ER-phagy.

(A) Mass spectrometry analysis identified a phosphorylation site on UBAC2 at serine 223. (B) Extracts of HeLa cells treated with TG (1 μM) for indicated time points were immunoprecipitated using phosphoserine/threonine/tyrosine polyclonal antibody and immunoblotted with UBAC2 antibody. (C) 293T cells transfected with vector of Flag-UBAC2 or its S223A mutant were treated with TG (1 μM) for 1 h in the presence of Baf A1 (0.2 μM). The lysates were subjected to coimmunoprecipitation and immunoblot analysis. (D) Lysates of 293T cells transfected with plasmids for Flag-UBAC2 and HA-UBAC2, were subjected to immunoprecipitation with anti-Flag and immunoblot analysis with anti-HA. (E) Extracts of HeLa cells cultured with increasing doses of TG for 1 h in the presence of Baf A1 (0.2 μM) were treated with 2 mM disuccinimidyl suberate (DSS) cross-linker and analyzed by immunoblotting. (F) Extracts of HeLa cells cultured with TG (1 μM) for indicated time points in the presence of Baf A1 (0.2 μM) were treated with 2 mM disuccinimidyl suberate (DSS) cross-linker and analyzed by immunoblotting. (G) Fluorescence microscopy analysis of HeLa cells with or without Baf A1 (0.2 μM) by labeling UBAC2 (green). Scale bar, 20 μm. (H) Quantitative analysis of the similar samples as (G) from three biologically independent experiments (20 cells scored per condition per experiment). (I) Fluorescence microscopy analysis of HeLa cells with or without TG (1 μM) for 1 h in the presence of Baf A 1 (0.2 μM) by labeling UBAC2 (green). Scale bar, 20 μm. (J) Quantitative analysis of the similar samples as (I) from three biologically independent experiments (20 cells scored per condition per experiment). (K) 293T cells transfected with plasmids encoding WT UBAC2 or its indicated mutants were treated TG (1 μM) for 1 h in the presence of Baf A1 (0.2 μM). The lysates were treated with 2 mM disuccinimidyl suberate (DSS) cross-linker and analyzed by immunoblotting. (L) HeLa cells transfected with vector of WT UBAC2 and its indicated mutants in the presence of Baf A 1 (0.2 μM) by labeling UBAC2 (green). Scale bar, 20 μm. (M) Quantitative analysis of the similar samples as (L) from three biologically independent experiments (20 cells scored per condition per experiment). (N) HeLa cells stably expressing the ER-phagy reporter were transfected with plasmids encoding WT UBAC2 or its indicated mutants and treated with doxycycline (Dox) (200 ng/mL) for 24 h. After incubation the cells with TG (1 μM) for 8 h, the lysates were harvested and subjected to immunoblot analysis. (O) Transmission electron microscopy of WT and *UBAC2* KO HeLa cells transfected with vector of Flag-UBAC2 or its indicated mutants. The ER was indicated by black arrowheads. Scale bar, 200 nm. Data information: For (B–G, I, K, L, N, O), one representative experiment out of three was shown. In (H, J, M), data are presented as the mean ± SEM of three independent biological experiments. The statistical significance of the difference was analyzed by unpaired two-tailed Student's *t* test, and the *P* values were shown. Source data are available online for this figure.

transcription of ER stress responsive genes as well as inflammatory genes in DSS mice (Fig. EV5G).

Previous reports have revealed that functional variants within the *UBAC2* genes are related to the increased risk of inflammatory diseases, including Behcet's disease, non-melanoma skin cancer, and Alzheimer's disease (Fei et al, 2009; Lai et al, 2022; Nan et al, 2011). We generated UBAC2 bearing R277C, F279S, or G293S mutant, which are variants assessed as somatic mutations. We found the interaction between GABARAP and UBAC2 harboring R277C, F279S, or G293S mutation was dramatically decreased (Fig. 7A). Using the ss-RFP-GFP-KDEL ER-phagy receptor assay, we observed the accumulation of cleaved RFP fragment induced by autophagy was abrogated in the cells expressing UBAC2 mutants (Fig. 7B,C). We transduced human UBAC2 mutants in AAV-delivered shRNA *Ubac2* knockdown mice to avoid the interference caused by endogenous mouse UBAC2 (Figs. 7D and EV5H), finding that DSS mice expressing mutated UBAC2 experienced apparent body weight loss compared with mice expressing WT UBAC2 (Fig. 7E). Consistently, the UBAC2 mutated DSS mice exhibited a remarkable increase of the shortening of colon lengths, and displayed much worse consistency and morphology of the colons with inflammatory infiltration (Fig. 7F–I). qPCR analysis of the intestines revealed that overexpression of the mutated form of UBAC2 increased the transcription of ER stress responsive genes and inflammatory genes in DSS mice (Fig. 7J). Collectively, our data demonstrate that UBAC2-mediated ER-phagy contribute to cellular homeostasis, thus preventing the occurrence and development of inflammatory diseases.

## Discussion

ER-phagy is crucial for the continuous renovation of ER to maintain its function and integrity (Ferro-Novick et al, 2021). ER undergoes degradation in response to virous intrinsic/extrinsic cellular stress, such as pathogen infection, nutrient deprivation, ER stress, or protein misfolding (Reggio et al, 2021). ER-phagy progress can be beneficial to ER-storage disorder associated diseases, including but not limited to osteogenesis imperfecta, α1-antitrypsin deficiency, and metaphyseal chondrodysplasia (Cameron et al, 2015; El-Gazzar et al, 2023; Reiterer et al, 2010). Thus, identification of novel ER-phagy receptors, along with discovery of molecular mechanisms of receptor activation, can provide ER-phagy-based therapeutic strategies to promote protein clearance in ER-storage disorders. In this study, we revealed that UBAC2 serves as a new ER-phagy receptor which harbors a conserved LIR motif to bind with GABARAP for mediating the ER turnover. UBAC2 underwent degradation upon nutrient deprivation or ER stress-induced autophagic conditions, while itself did not influence the autophagic flux. The interaction between UBAC2 and GABARAP, the degradation of UBAC2, as well as ER turnover was abrogated when the LIR motif was mutated. In *Arabidopsis*, UBAC2 proteins associate with ATG8-interacting ATI3 proteins and function in plant stress responses (Zhou et al, 2018). Our results indicate that UBAC2 can be a eukaryotic conserved ER regulator for maintaining ER homeostasis.

Recent studies indicate that the activity of ER-phagy is regulated by the post-translational modifications of ER-phagy receptors. E3 ubiquitin ligase AMFR catalyzes the ubiquitination of FAM134B within its reticulon homology domain (RHD) enhances receptor clustering and binding to matured LC3B, thus prompting the progress of ER-phagy (González et al, 2023). ER resident UFMylation is responsible for ER-phagy to suppress the UPR via IRE1a. The UFL1 ligase is recruited to the ER surface by DDRGK1 (also named as UFBP1) to catalyze the UFMylation of RPN1 and RPL26 and preferentially delivers ER sheets for autophagic degradation (Liang et al, 2020). In our MS data generated from purified UBAC2 under ER stress condition, we identified a specific phosphorylated peptide of UBAC2. Further study demonstrated that the phosphorylation of UBAC2 at the S223 is indispensable for the sequestration of UBAC2-resided ER for degradation. We uncovered that MARK2 interacted with UBAC2 and catalyzed the phosphorylation of UBAC2 at the S223. MARK2 is a serine/threonine kinase involved in the modulation of microtubule stability (Hurov et al, 2004; Suzuki et al, 2004). Previous research has indicated that upon protein misfolding stress, MARK2 serves as

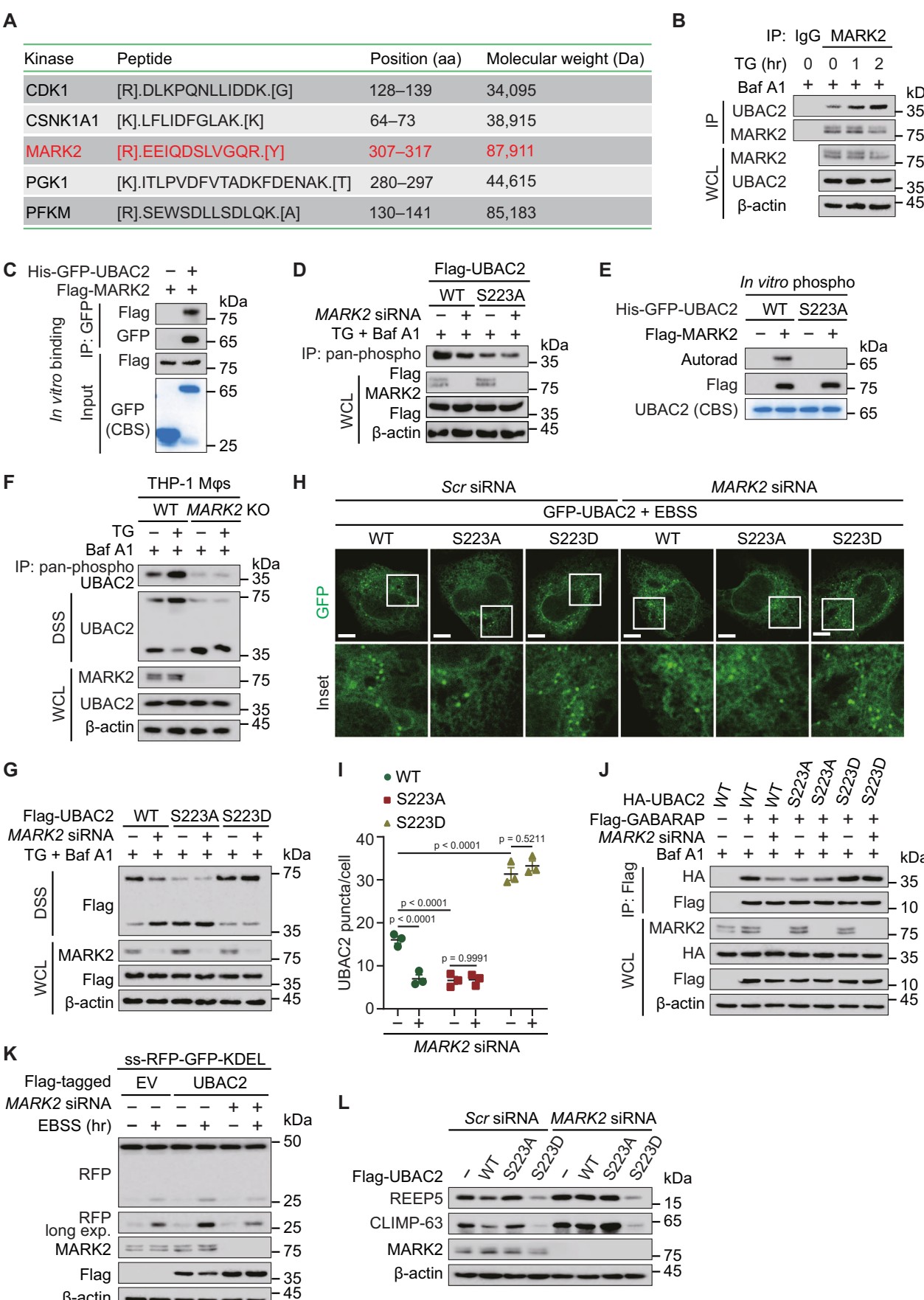

**Figure 5. MARK2 catalyzes the phosphorylation of UBAC2 at S223.**

(A) Lysates prepared from HeLa cells were immunoprecipitated with antibody against UBAC2. The precipitated proteins were subjected to mass spectrometry analysis and the identified peptides of kinase were presented in the table. This experiment was performed once with two replicates per condition. (B) Lysates of HeLa cells treated with TG (1 μM) in the presence of Baf A1 (0.2 μM) were subjected to immunoprecipitation and immunoblot analysis. (C) Flag-MARK2 was expressed in 293T cells, purified using Flag affinity column, and eluted with Flag peptide. The purity of protein was analyzed by SDS-PAGE and visualized by Coomassie Blue staining (CBS). His-GFP-UBAC2 purified from *E. coli* were incubated with Flag-MARK2 in reaction buffer in vitro, and then immunoprecipitated and analyzed by immunoblotting. (D) HeLa cells were transfected with scramble or *MARK2*-specific siRNA and treated with TG (1 μM) for 1 h in the presence of Baf A1 (0.2 μM). The lysates were subjected to immunoprecipitation and immunoblot analysis. (E) Recombinant proteins for WT UBAC2 and UBAC2 S223A purified from *E. coli* were incubated with purified MARK2 from 293T cells by IP in kinase buffer. In vitro UBAC2 S223 phosphorylation by MARK2 was analyzed by autoradiography. (F) WT or *MARK2* KO THP-1 Mφs were treated with TG (1 μM) for 1 h in the presence or presence of Baf A1 (0.2 μM). The lysates were treated with 2 mM disuccinimidyl suberate (DSS) cross-linker or immunoprecipitated using phosphoserine/threonine/tyrosine polyclonal antibody, and detected by immunoblot. (G) 293T cells were transfected with scramble or *MARK2*-specific siRNA and then transfected with vector of WT or S223A UBAC2. After incubation the cells with TG (1 μM) for 1 h in the presence of Baf A1 (0.2 μM), the lysates were treated with 2 mM disuccinimidyl suberate (DSS) cross-linker and analyzed by immunoblotting. (H) HeLa cells were transfected with scramble or *MARK2*-specific siRNA alongside with vector of WT or mutated UBAC2. The cells were cultured in EBSS for 3 h and observed by fluorescence microscopy. Scale bar, 20 μm. (I) Quantitative analysis of the similar samples as (H) from three biologically independent experiments (20 cells scored per condition per experiment). (J) 293T cells were transfected with *MARK2*-specific or scramble siRNA and then transfected with vector of Flag-GABARAP, HA-tagged WT or mutated UBAC2. After incubation the cells with Baf A1 (0.2 μM), the lysates were harvested for immunoprecipitation and immunoblot analysis. (K) HeLa cells stably expressing the ER-phagy reporter were transfected with scramble or *MARK2*-specific siRNA and then transfected with vector of Flag-UBAC2. The cells were cultured in EBSS for 3 h were harvested for immunoblot analysis. (L) HeLa cells were transfected with plasmids expressing Flag-tagged WT UBAC2 or its indicated point mutants. The cells were cultured in EBSS for 3 h and the lysates were harvested for immunoblotting analysis. Data information: For (B–H, J–L), one representative experiment out of three was shown. In (I), data are presented as the mean ± SEM of three independent biological experiments. The statistical significance of the difference was analyzed by unpaired two-tailed Student's *t* test, and the *P* values were shown. Source data are available online for this figure.

a direct kinase for eIF2α (Lu et al, 2021). Phosphorylation promotes the formation of UBAC2 dimers, thus enhances the association between with UBAC2 and GABARAP to deliver ER for autophagic degradation. Our data revealed that MARK2 as an ER stress responsive kinase phosphorylates UBAC2 at S223, which provides a selective signal for targeting ER in responding to ER stress. The detailed responsive mechanism of MARK2 upon ER stress will be an attractive issue to be explored in the future.

ER homeostasis and quality control orchestrating the cellular fate and function, are at the foundation of almost all biological activities (Koksal et al, 2021). The dysregulation of ER homeostasis is emerging as a possible driver of a variety of inflammatory diseases, such as Crohn disease and type 2 diabetes (Ren et al, 2021). To surveil ER quality, the UPR is initiated through the activation of ATF6, PERK, and IRE1α (Hetz et al, 2020a). ER stress triggers the expression of pro-inflammatory cytokine IL-6 through NOD1/2-dependent signaling pathway. *Brucella abortus* injects the type IV secretion system effector protein VceC into host cells to induce ER stress (Keestra-Gounder et al, 2016). The interplay between NOD1/2-directed pro-inflammatory responses and IRE1α/TRAF2 signaling pathway indicated that ER-stress contributes to the innate immune regulation. Despite the importance of ER-phagy in inflammatory diseases, the underlying mechanisms ER-phagy in the progression of disease remain elusive. *UBAC2* depletion increased the inflammatory responses through modulating ER-phagy. The UPR was aggravated in *UBAC2*-deficienct cells or cells expressing UBAC2 with the LIR mutant. In mice expressing disease-associated human UBAC2 mutants, the ER-phagy was decreased, while the inflammatory responses and DSS-induced colitis was incensed. Our data suggest that UBAC2 servers as a guardian for ER-phagy and ER turnover, thereby suppressing the inflammatory responses to restrain the immunity in a balance. Since *UBAC2* is a high frequency mutation gene associated with Behcet's disease, skin, and bladder cancer, the immune dysregulation associated with ER-phagy can be a promising therapy target for further clinical diagnosis and treatment.

Taken together, UBAC2 functions as a newly identified ER-phagy receptor, which specifically interacts with GABARAP using its conserved LIR motif. Under ER stress or starvation induced autophagy conditions, MARK2 catalyzes the phosphorylation of UBAC2 at S223 to promote the dimerization of UBAC2. Through UBAC2 dimerization, the UBAC2-resided ER can be targeted and sequestered by autophagosomes for selective degradation. UBAC2 suppresses the inflammatory responses through the UPR in an ER-phagy-dependent manner (Fig. EV5I). Our findings provide novel evidence to dissect the pathogenic mechanism of UBAC2 in inflammatory diseases and expand our understanding of therapeutic application based on ER-phagy.

# Methods

## Mice

Wild-type C57BL/6 mice, both male and female, between the age of 8 to 12 weeks, were acquired from Guangzhou Medical Laboratory Animal Center of China. Mice were housed in a specific pathogen-free facility with a temperature of 20–26 °C, humidity between 40 and 70%, under 12 h light/dark cycle at Sun Yat-sen University. All animal experimental protocols were approved by the Institutional Animal Care and Use Committee (IACUC) of Sun Yat-sen University [Authorization number: SYXK (YUE) 2023-0313, Guangzhou, China].

## Cell culture and reagents

HEK293T (#GNHu17) and HeLa (#TCHu187) cells obtained from National Collection of Authenticated Cell Cultures (Shanghai, China), were cultured in DMEM medium (Corning, cat. 10-013-CV) with 10% fetal bovine serum (FBS) [Gibco, cat. 10099141] and 1% glutamine (Gibco, cat. 35050061). The human monocytic THP-1 cell line (#TIB-202) purchased from American Type Culture Collection (ATCC) was cultured in RPMI 1640 medium (Gibco, cat. C22400500BT) supplemented with 10% FBS and 1% glutamine. THP-1 cells were differentiated into macrophages cultured with RPMI 1640 containing 100 ng/mL phorbol-12-myristate-13-acetate

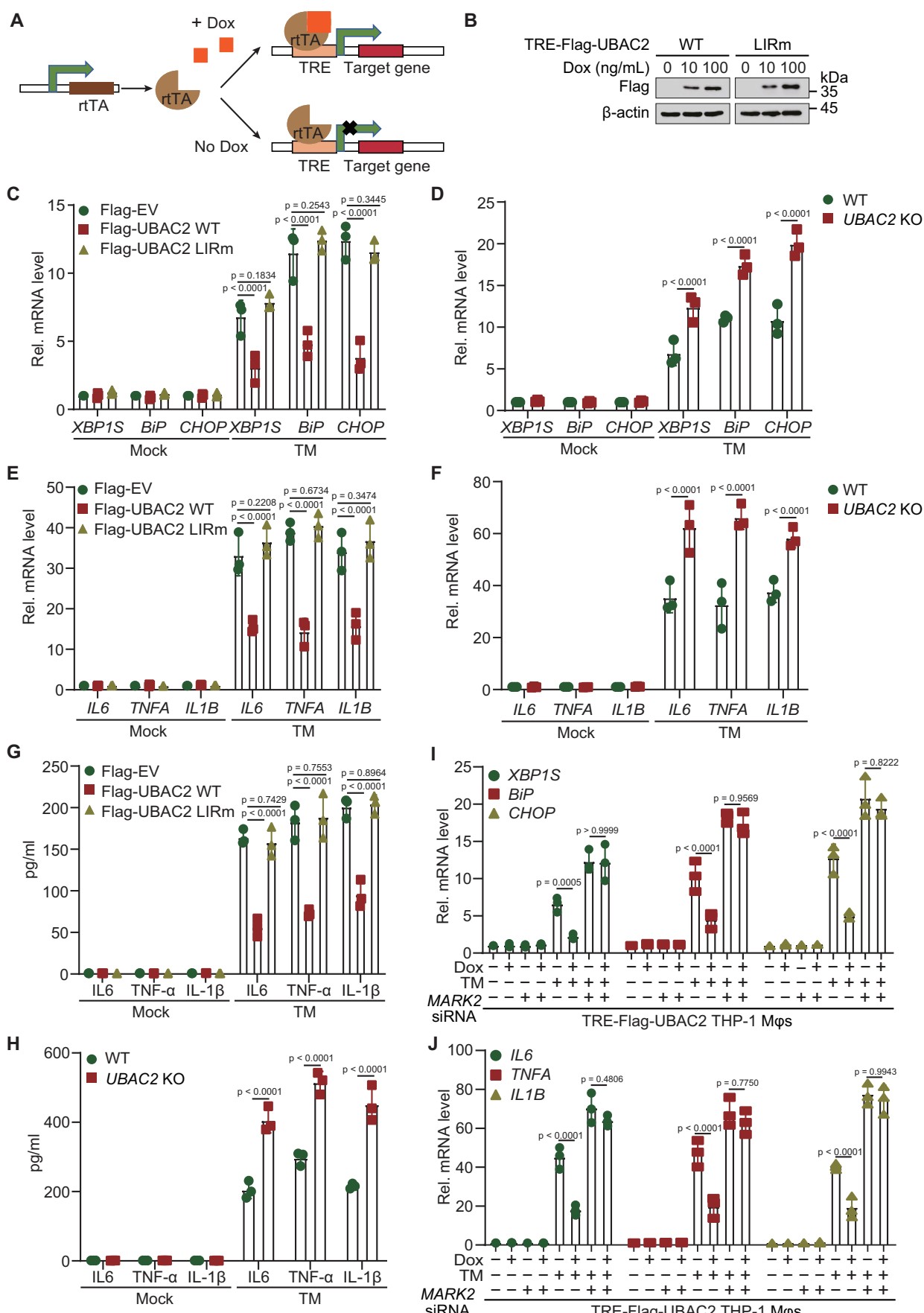

◀ **Figure 6. MARK2-UBAC2 axis suppresses UPR and inflammatory responses through ER-phagy.**

(A) Schematic of gene regulation in the BDTM Tet-On Systems. The transcription is turned on by the reverse tetracycline-responsive transcriptional activator (rtTA), which binds to the tetracycline-responsive element (TRE) promoter and activates transcription in the presence of doxycycline (Dox). (B) Lysates of Flag-tagged WT or LIRm UBAC2 inducible THP-1 cells incubated with Dox as indicated concentration for 12 h were analyzed by immunoblot. (C) qPCR analysis of ER-stress inducible transcripts in WT or LIRm UBAC2 inducible THP-1 cells were incubated with Dox (200 ng/mL) for overnight and treated with TM (5 µg/mL) for 6 h. (D) qPCR analysis of ER-stress inducible transcripts in WT and *UBAC2* KO THP-1 cells with TM (5 µg/mL) treatment for 6 h. (E) qPCR analysis of inflammatory genes in WT or LIRm UBAC2 inducible THP-1 cells were incubated with Dox (200 ng/mL) for overnight and treated with TM (5 µg/mL) for 12 h. (F) qPCR analysis of inflammatory genes in WT and *UBAC2* KO THP-1 cells with TM (5 µg/mL) treatment for 12 h. (G) ELISA of inflammatory cytokines in WT or LIRm UBAC2 inducible THP-1 cells were incubated with Dox (200 ng/mL) for overnight and treated with TM (5 µg/mL) for 24 h. (H) ELISA of inflammatory cytokines in WT and *UBAC2* KO THP-1 cells with TM (5 µg/mL) treatment for 24 h. (I, J) Flag-UBAC2 inducible THP-1 cells incubated with or without Dox (200 ng/mL) for overnight were transfected with *MARK2*-specific or scramble siRNA. After incubation with TM (5 µg/mL) for 12 h, the expression of ER-stress inducible transcripts (I) and inflammatory genes (J) was analyzed by qPCR. Data information: For (B), one representative experiment out of three was shown. In (C–J), data are presented as the mean ± SEM of three independent biological experiments. The statistical significance of the difference was analyzed by unpaired two-tailed Student's *t* test, and the *P* values were shown. Source data are available online for this figure.

(PMA) [Sigma-Aldrich, cat. P1585] for overnight, and then the macrophages had a resting period containing 24 h before stimulated. HT-29 cells (#SCSP-5032) obtained from National Collection of Authenticated Cell Cultures were cultured in McCoy's 5a Medium (Gibco, cat. 16600082) supplemented with 10% FBS. To induce starvation, cells were washed with phosphate-buffered saline (PBS) and incubated in EBSS (Gibco). MG132 (cat. C-2211) and Bafilomycin A1 (cat. 19-148) were purchased from Sigma-Aldrich. CPA (cat. ab120398) and TG (cat. ab120286) were from Abcam. Carfilzomib (cat. HY-10455), tunicamycin (cat. HY-A0098) and PEP005 (cat. HY-B0719) were purchased from MedChemExpress (MCE).

## Antibodies

Anti-Beclin-1 (3738), anti-p62 (8025), anti-MARK2 (9118S), anti-PKC-δ (2058S) and antibody to PKC-δ phosphorylated at T505 (93292S) were purchased from Cell Signaling Technology. Horseradish peroxidase (HRP)-anti-Flag (M2) (A8592) and anti-β-actin (A1978) were purchased from Sigma-Aldrich. HRP-anti-hemagglutinin (clone 3F10) was purchased from Roche Applied Science. Anti-RFP (Ab124754), anti-LAMP1 (ab25630), anti-WIPI2 (ab105459) and antibody to MARK2 phosphorylated at T595 (ab34751) were purchased from Abcam. Phosphoserine/threonine/tyrosine polyclonal antibody (#61-8300) were obtained from Invitrogen. Anti-GABARAP (18723-1-AP), anti-GFP (66002-1-Ig), anti-UBAC2 (25122-1-AP), anti-ATG16L1 (67943-1-Ig), anti-CHOP (66741-1-AP), anti-BiP (66574-1-AP), anti-XBP1S (24868-1-AP), anti-CCPG1 (13861-1-AP), anti-RTN3 (12055-2-AP), anti-SEC62 (28693-1-AP), anti-TEX264 (25858-1-AP), anti-ATL3 (16921-1-AP) and anti-REEP5 (14643-1-AP) were purchased from Proteintech Group. Anti-CLIMP-63 (ALX-804-604) was obtained from Enzo Life Sciences.

## Plasmids and siRNA transfection

Constructs coding for UBAC2 was cloned in the pcDNA3.1 vector for transient expression and into the FG-EH-DEST (provided by Xiaofeng Qin laboratory) for retroviral expression. Site-directed mutagenesis was performed with the QuickChange Lightning Kit (210519-5; Agilent Technologies) according to the manufacturer's instructions. Chemically synthesized 21-nucleotide siRNA duplexes were obtained from Sangon Biotech and transfected using Lipofectamine RNAiMAX (Invitrogen, cat. 13778030) according

to the manufacturer's instructions. RNA oligonucleotides used in this study are as follows:

*Scr*: 5′-GUUAUCGCAACGUGUCACGUA-3′;
*UBAC2* siRNA #1: 5′-GCUUUCAAGUGGAGGGAAGUU-3′;
*UBAC2* siRNA #2: 5′-GCAGCUGAUGUUCUCUCAGUU-3′;
*UBAC2* siRNA #3: 5′-GCUUCAAACAAUGACCUCAAU-3′;
*MARK2* siRNA #1: 5′-GUGGCGGAGAGGUAUUUGAUU-3′;
*MARK2* siRNA #2: 5′-AGAUGAUGAACUAAAGCCUUA-3′;
*MARK2* siRNA #3: 5′-CCUGAAUGAACCUGAAAGCAA-3′;
*FAM134B* siRNA #1: 5′-AGCUAUCAAAGACCAGUUATT-3′;
*FAM134B* siRNA #2: 5′-GCUCAGCCACUGUAUUGCAGA-3′;
*FAM134B* siRNA #3: 5′-CCUCUGAACAGUGACCAAATT-3′.

## Generation of inducible expression and knockout cell lines

For UBAC2-inducible expression, lentiviral particles were produced by transfecting HEK293T cells with pL-Teton3G-iZ-UBAC2-WT/UBAC2-LIRm and the Δ8.9 and VSVG lentivirus expression system. THP-1 cells were infected by incubation with lentivirus-containing supernatant for 48 h. Cells were treated with doxycycline to induce expression of Flag-tagged UBAC2 and UBAC2 LIRm. HeLa cells were infected by incubation with retrovirus-containing supernatant for 48 h. For *UBAC2* KO, *BECN1* KO, and *MARK2* KO cells, target sequences were cloned into pLentiCRISPRv2 by cutted with *Bsm*BI. The target sequences used were as follows:

*UBAC2* target #1: 5′-CAAAACGTAAACTATCAGGG-3′;
*UBAC2* target #2: 5′-ACGTAAACTATCAGGGCGGT-3′;
*UBAC2* target #3: 5′-AGTCGTTCTTGACTGCGTGA-3′;
*BECN1* target #1: 5′-GGGTCTCTCCTGGTTTCGCC-3′;
*BECN1* target #2: 5′-GGTCTCTCCTGGTTTCGCCT-3′;
*BECN1* target #3: 5′-ATTTATTGAAACTCCTCGCC-3′;
*MARK2* target #1: 5′-GGAGCCGGTAGTTTCCAATG-3′;
*MARK2* target #2: 5′-GACCGAACTAACTTCCCCCG-3′;
*MARK2* target #3: 5′-ACCCCGTTGAGAGAGAGCCG-3′.

## Real-time quantitative PCR (qPCR)

Total RNA was extracted from cells using the TRIzol reagent (Invitrogen, cat. 15596018) and cDNA was generated with HiScript® II Q RT SuperMix for qPCR (+gDNA wiper) (Vazyme, cat. R223-01). qPCR analysis was performed using the 2 × PolarSignal SYBR Green mix Tag (with Tli RNaseH) (MIKX, cat.

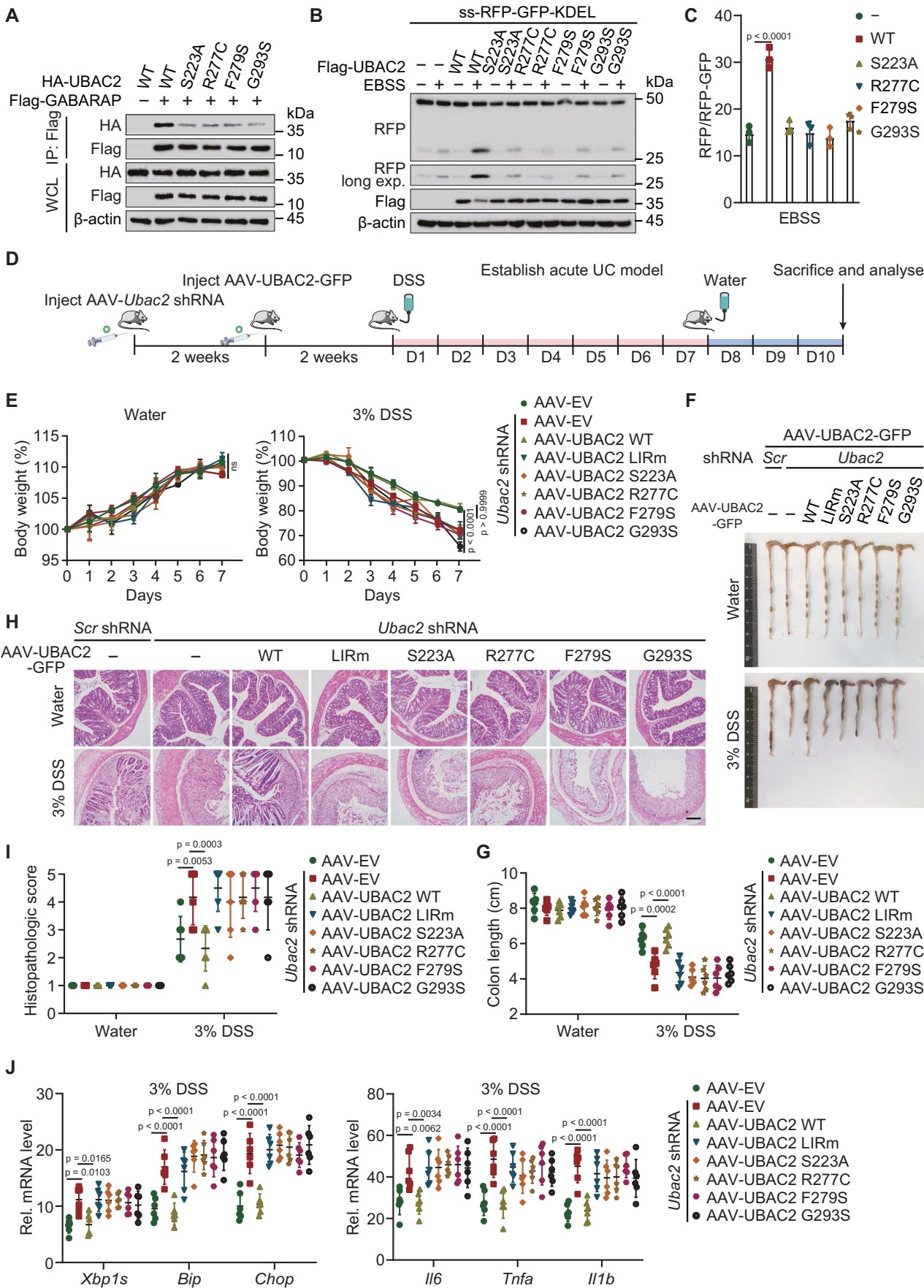

◄ **Figure 7. UBAC2 restrains DSS-induced colitis in mice through ER-phagy.**

(A) Lysates of 293T cells transfected with plasmids of Flag-GABARAP, alongside with HA-UBAC2 and its indicated mutants were immunoprecipitated with α-Flag M2 beads and immunoblotted with anti-HA. (B) HeLa cells stably expressing the ER-phagy reporter were transfected with plasmids encoding WT UBAC2 or its indicated mutants and treated with doxycycline for 24 h. After incubation the cells with EBSS for 3 h, the lysates were harvested and subjected to immunoblot analysis. (C) The band intensities of RFP and RFP-GFP were quantified and the ratios of RFP/RFP-GFP in the similar sample as (B). (D) Flow chart of the experimental design of acute ulcerative colitis (UC) model. (E) Changes in body weight (percentage of original body weight) over time (days) in water- or DSS-treated mice expressing WT and mutated UBAC2 (n = 6 independent biological mice per group). (F) Representative macroscopic features of colons from water- or DSS-treated mice expressing WT and mutated UBAC2. (G) Colon length (n = 6 independent biological mice per group) of water- or DSS-treated mice expressing WT and mutated UBAC2. (H, I) Representative images (H) of H&E staining of colon sections and the histopathologic scores (I) from water- or DSS-treated mice expressing WT and mutated UBAC2 (n = 6 independent biological mice per group). Scale bar, 100 μm. (J) qPCR analysis of ER-stress inducible transcripts (left) and inflammatory genes (right) of intestine homogenates from DSS mice expressing WT and mutated UBAC2. Data information: For (A, B), one representative experiment out of three was shown. In (C), data are presented as the mean ± SEM of three independent biological experiments. In (E), data are presented as mean ± SEM (n = 6 independent biological mice per group). In (G, I, J), data are presented as mean ± SD (n = 6 independent biological mice per group). The statistical significance of the difference was analyzed by unpaired two-tailed Student's t test (C, G, I, J) or two-way ANOVA with Bonferroni's multiple comparisons test (E), and the P values were shown. Source data are available online for this figure.

MKG900-10) and the data were normalized to *GAPDH* or *Actin* expression. Primer sequences were:

Human *UBAC2*: 5′-CCAGTGGGCTCTACAAGGCG-3′, 5′-CAAAACCCAGGAACCCAGCA-3′;

Human *IL1B*: 5′-AGCTACGAATCTCCGACCAC-3′, 5′-CGTTATCCCATGTGTCGAAGAA-3′;

Human *IL6*: 5′-ACTCACCTCTTCAGAACGAATTG-3′, 5′-CCATCTTTGGAAGGTTCAGGTTG-3′;

Human *TNFA*: 5′-CCTCTCTCTAATCAGCCCTCTG-3′, 5′-GAGGACCTGGGAGTAGATGAG-3′;

Human *XBP1S*: 5′-CCCTCCAGAACATCTCCCCAT-3′, 5′-ACATGACTGGGTCCAAGTTGT-3′;

Human *CHOP*: 5′-GGAAACAGAGTGGTCATTCCC-3′, 5′-CTGCTTGAGCCGTTCATTCTC-3′;

Human *BiP*: 5′-GAGTTCTTCAATGGCAAGGA-3′, 5′-CCAGTCAGATCAAATGTACCC-3′;

Human *GAPDH*: 5′-CGGAGTCAACGGATTTGGTC-3′, 5′-GACAAGCTTCCCGTTCTCAG-3′;

Mouse *Il1b*: 5′-AAAGATGAAGGGCTGCTTCCA-3′, 5′-CTGCGAGATTTGAAGCTGGAT-3′;

Mouse *Il6*: 5′-AGCTGGAGTCACAGAAGGAG-3′, 5′-AGGCATAACGCACTAGGTTT-3′;

Mouse *Tnfa*: 5′-GTCAGGTTGCCTCTGTCTCA-3′, 5′-TCAGGGAAGAATCTGGAAAG-3′;

Mouse *Xbp1s*: 5′-CTGAGTCCGCAGCAGGTG-3′, 5′-TGGCTGGATGAAAGCAGGTT-3′;

Mouse *Chop*: 5′-GTCCCTAGCTTGGCTGACAGA-3′, 5′-TGGAGAGCGAGGGCTTTG-3′;

Mouse *Bip*: 5′-GAGTTCTTCAATGGCAAGGA-3′, 5′-CCAGTCAGATCAAATGTACCC-3′;

Mouse *Actin*: 5′-CTTCTTGGGTATGGAATCCT-3′, 5′-GGAGCAATGATCTTGATCTT-3′.

## Immunoprecipitation and immunoblot analysis

The whole-cell extracts collected from cells with indicated treatments were incubated with appropriate antibodies plus protein A/G beads (Thermo Scientific, cat. 20424) at 4 °C for overnight. For immunoprecipitation with anti-Flag, α-Flag M2 beads (Millipore, cat. A2220) were used. The beads were washed 5 times with low-salt lysis buffer (50 mM HEPES, 150 mM NaCl, 1 mM EDTA, 10% glycerol, 1.5 mM MgCl$_2$, and 1% Triton X-100) and eluted with 2 × SDS Loading Buffer (FD Biotechnology, cat. FD006) for

immunoblotting. Proteins were transferred to PVDF membrane (Bio-Rad, cat. 1620177) and incubated with indicated antibodies. Protein bands were imaged with Immobilon Western Chemiluminescent HRP Substrate (Millipore, cat. WBKLS) in a ChemiDoc XRS+ System (Bio-Rad Laboratories) using the Image Lab software.

## In vitro recombinant protein purification and phosphorylation assay

The expressing plasmids encoding His-tagged WT or UBAC2 mutants conjugated GFP were transformed into *E. coli* (strain BL21), and the recombinant proteins were purified as previous described (Cai et al, 2023). Recombinant proteins for UBAC2 and its mutant were incubated with purified Flag-MARK2 from 293T cells by IP in the reaction buffer (25 mM Tris-HCl pH 7.5, 10 mM MgCl$_2$, 50 μM ATP, 1 mM DTT, and 1 μCi [γ-$^{32}$P] ATP) in a 25 μL total volume at 30 °C for 0.5 h. The reaction was terminated with 6 × SDS loading buffer. After incubation at 100 °C for 5 min, the reaction products were separated on 12% SDS-PAGE and stained using Coomassie Brilliant Blue R 250. The phosphorylation signals were detected by a Typhoon 9410 phosphor imager.

## Biotin labeling with ERM-PhastID in live cells

The pcDNA3.1-PhastID was provided by Dr. Feng Liu (Sun Yat-sen University). A endoplasmic reticulum membrane (ERM) anchor derived from cytochrome P450 containing amino acids named as C1(1–27) (Branon et al, 2018) was expressed by fusion with PhastID protein at its N terminal. For labeling of the proteins in ERM-facing cytosol in live cells, we initiated biotin (50 μM) labeling for 1 h following 2 h TG treatment. Labeling was stopped after the desired time by transferring the cells to ice and washing five times with ice-cold PBS. For negative controls, we omitted exogenous biotin.

## Mass spectrometry analysis

Total lysates prepared from HeLa cells were immunoprecipitated with UBAC2. The eluates were resolved by SDS-PAGE. and then stained with Coomassie Blue to excise the entire lane. Afterward, the samples were digested with trypsin and the peptides were chromatographed through the Easy-nLC 1000 system (Thermo

Fisher) at Applied Protein Technology Corporation (Shanghai, China). LC-MS/MS identification of peptide mixtures was performed once with two replicates per condition using Q Exactive mass spectrometer (Thermo Fisher). RAW files generated by the spectrometer was processed in Proteome Discoverer (version 1.4) software with searching the library of Uniprot homo sapiens (uniprot_Homo_sapiens_188433_20200217.fasta) for protein identification. Trypsin was specified as the proteolytic enzyme, with up to two missed cleavage sites allowed. Carbamidomethylation (C) was set as the fixed modification. Oxidation (M) and phosphorylation (S, T, Y) were set as the variable modifications. The precursor mass tolerance was set to 20 ppm and the fragment ion tolerance at 0.1 Da. Proteins were identified based on at least one unique peptide. Protein and peptide identification confidence threshold were set to an FDR of 1%. The tandem mass spectra of matched phosphorylated peptides were further checked manually for their validity.

## Confocal microscope

Cells cultured on glass-bottom culture dishes (Nest Scientific) were fixed with 4% paraformaldehyde for 10 min and permeabilized in methyl alcohol for 30 min at $-20\,°C$. The samples washed with PBS for 3 times were blocked in 5% goat serum for 1 h and incubated with primary antibodies diluted in 5% goat serum for overnight. After washing with PBS for 3 times, the cells fluorescently labeled with secondary antibody for 1 h. Confocal images were detected using a super-resolution confocal microscope (TCS SP8 STED 3X; Leica) equipped with $100 \times 1.40$ NA oil objectives and processed for gamma adjustments using Leica AS Lite or ImageJ software (National Institutes of Health). The data plotted on the line intensity plots were generated using the Plot Profile function of ImageJ on a single plane z-stack of confocal images.

## Transmission electron microscopy

HeLa cells were grown on glass-bottom plates before stimulation with EBSS. Samples were fixed, sectioned, stained, and coated by the Core Facility of School of Life Sciences, Sun Yat-sen University. The images were examined at 120 kV in JEM-1400Flash transmission electron microscope (JEOL) coupled to a high-sensitivity sCMOS camera.

## AAV treatments

Construct coding for wide-type UBAC2 was cloned into the pAAV-MCS-hrGFP, and plasmid mutants of UBAC2 were generated by site-directed mutagenesis using the Mut Express® II Fast Mutagenesis Kit V2 (Vazyme, cat. C214-01). AAVsh-mTurquoise2 vectors expressing short hairpin RNA (shRNA) of scramble (target sequence: 5′-TTCTCCGAACGTGTCACGTTT-3′) and *Ubac2* (target sequence: 5′-GCAGCTGATGCTCTCTCAGTT-3′) were generated according to standard molecular biology technique. The above empty vectors were kindly gifts provided by Dr. Yuanchu She in Dr. Sheng-Jian Ji's laboratory (Southern University of Science and Technology). The administration procedures were performed according to previous studies (Fang et al, 2019; Zhou et al, 2022). Briefly, $5 \times 10^{10}$ physical particles of AAV in 100 μL of PBS were

injected into the tail veins of 8-week-old C57BL/6 mice. Small intestine and colon were removed for GFP or UBAC2 expression evaluation with western blots.

## Dextran sodium sulfate-induced colitis

Acute dextran sodium sulfate (DSS) colitis was induced as previously described (Chassaing et al, 2014; Zhao et al, 2020). Acute colitis was induced in 8–10-week-old male and female mice by oral administration of 3% or 5% DSS (36–50 kDa; Yeasen, cat. 60316ES60) in the drinking water during 7 days followed by 2–3 days on normal drinking water. All mice were killed at day 10. The body weight of the mice was monitored daily or every other day in case of the acute model. The acute DSS experiments were repeated three times whereby results could be replicated each time. No randomization was applied before start of the experiments.

## Histological assessment

Intestines from DSS mice were dissected, fixed in 4% paraformaldehyde for more than 24 h, and embedded in paraffin. The sections (thickness, 6 μm) were stained with hematoxylin & eosin Y (Servicebio, China). Hematoxylin & eosin-stained sections were analyzed by microscope (DMi8; Leica).

## ELISA

The concentrations of cytokines in culture supernatants were measured by human IL6 ELISA Kit (BD Biosciences, cat. 555220), human TNF-α ELISA Kit (BD Biosciences, cat. 555212), and human IL-1β ELISA Kit (BD Biosciences, cat. 557953).

## Statistical analyses

Data from three or more experiments were displayed as mean ± SEM unless otherwise indicated, and data of animal experiments are represented as mean ± SD of indicated numbers of mice. Student's $t$-test was used for all statistical analyses with the GraphPad Prism 8 software. Differences between two groups were considered significant when $P$ value was less than 0.05.

# Data availability

All data have been included in the manuscript, figures, extended view information, appendix figures and the Source Data files. The mass spectrometry proteomics data of biotin labeling with ERM-PhastID have been deposited to the ProteomeXchange Consortium via the PRIDE partner repository with the dataset identifier PXD052877. The mass spectrometry proteomics data of UBAC2 phosphorylation have been deposited to the ProteomeXchange Consortium via the PRIDE partner repository with the dataset identifier PXD052418.

The source data of this paper are collected in the following database record: biostudies:S-SCDT-10_1038-S44318-024-00232-z.

## Peer review information

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

## Acknowledgements

This work was supported by the National Natural Science Foundation of China (32170876 to SJ, 31970700 to SJ, and 22370922 to JC), the National Key R&D Program of China (2020YFA0908700 to JC), Guangdong Basic and Applied Basic Research Foundation (2020B1515120090 to JC), the Fundamental Research Funds for the Central Universities, Sun Yat-sen University (23lgbj012 to SJ and 23yxqntd001 to JC), the Guangdong Province Excellent Youth Team Project (2024B1515040009 to JC and SJ), the Natural Science Foundation of Guangdong Province (2022A1515012223 to YQ), and Science and Technology Program of Guangzhou, China (202201020430 to YQ).

## Author contributions

**Xing He**: Conceptualization; Data curation; Software; Formal analysis; Validation; Investigation; Visualization; Methodology; Writing—original draft; Writing—review and editing. **Haowei He**: Investigation; Methodology. **Zitong Hou**: Investigation; Methodology. **Zheyu Wang**: Validation; Methodology. **Qinglin Shi**: Validation; Methodology. **Tao Zhou**: Software; Visualization. **Yaoxing Wu**: Validation; Methodology. **Yunfei Qin**: Validation; Methodology. **Jun Wang**: Software; Formal analysis. **Zhe Cai**: Software; Formal analysis. **Jun Cui**: Resources; Funding acquisition. **Shouheng Jin**: Conceptualization; Resources; Data curation; Software; Formal analysis; Supervision; Funding acquisition; Validation; Investigation; Visualization; Methodology; Writing—original draft; Project administration; Writing—review and editing.

Source data underlying figure panels in this paper may have individual authorship assigned. Where available, figure panel/source data authorship is listed in the following database record: biostudies:S-SCDT-10_1038-S44318-024-00232-z.

## Disclosure and competing interests statement

The authors declare no competing interests.

# Expanded View Figures

**Figure EV1.  UBAC2 targets GABARAP vis its LIR.**

(A) The quantification of UBAC2 protein abundances in the similar samples as Fig. 2A from three independent experiments performed in duplicate. (B, C) HEK293T cells transfected with plasmids encoding HA-UBAC2 and Flag-tagged ATG8 family members were treated with TG (1 μM) for 1 h (B) or EBSS for 3 h (C) in the presence Baf A1 (0.2 μM). The lysates were immunoprecipitated with anti-Flag and immunoblotted with anti-HA. (D) 293T cells transfected with vectors encoding HA-UBAC2 and Flag-GABARAP were treated with TG (1 μM) for indicated time points in the presence of Baf A1 (0.2 μM), followed by immunoprecipitation with anti-Flag beads and immunoblot analysis with anti-HA. (E) 293T cells transfected with vectors encoding HA-UBAC2 and Flag-GABARAP were cultured in EBSS for indicated time points with the existence of Baf A1 (0.2 μM), followed by immunoprecipitation with anti-Flag beads and immunoblot analysis with anti-HA. (F, G) The quantification of UBAC2 protein abundances in the similar samples as Fig. 2B (F) and Fig. 2C (G) from three independent experiments performed in duplicate. (H) Domain organization of human UBAC2 and alignment of UBAC2 sequences of different species. (I, J) The quantification of UBAC2 protein abundances in the similar samples as Fig. 2G (I) and Fig. 2K (J) from three independent experiments performed in duplicate. (K) Immunoprecipitation and immunoblot analysis of 293T cells transfected with vectors encoding HA-UBAC2 and Flag-GABARAP as well as its indicated mutants. Data information: For (B–E, K), one representative experiment out of three was shown. In (A, F, G, I, J), data are presented as the mean ± SEM of three independent biological experiments. The statistical significance of the difference was analyzed by unpaired two-tailed Student's *t* test, and the *P* values were shown. Source data are available online for this figure.

▶

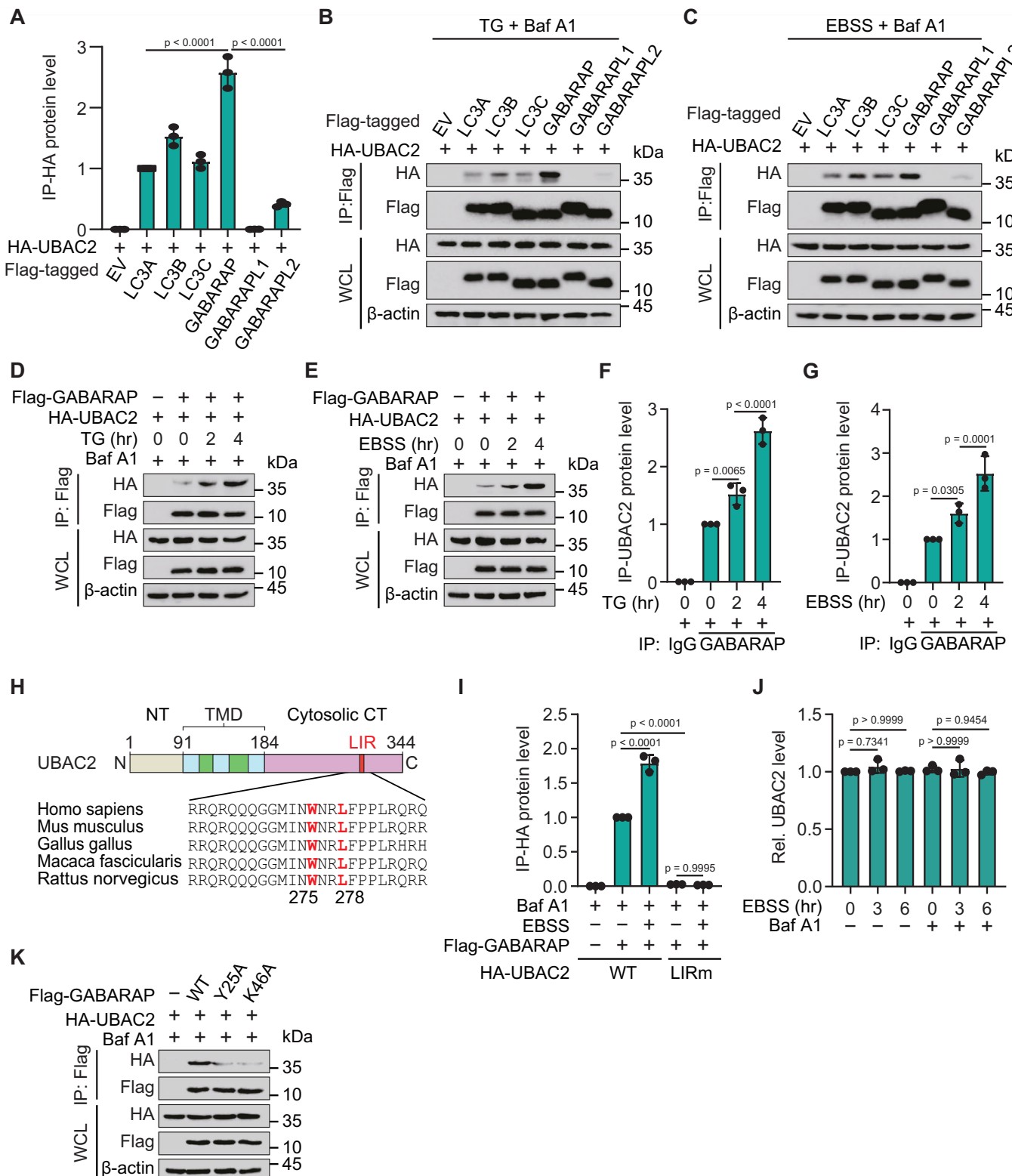

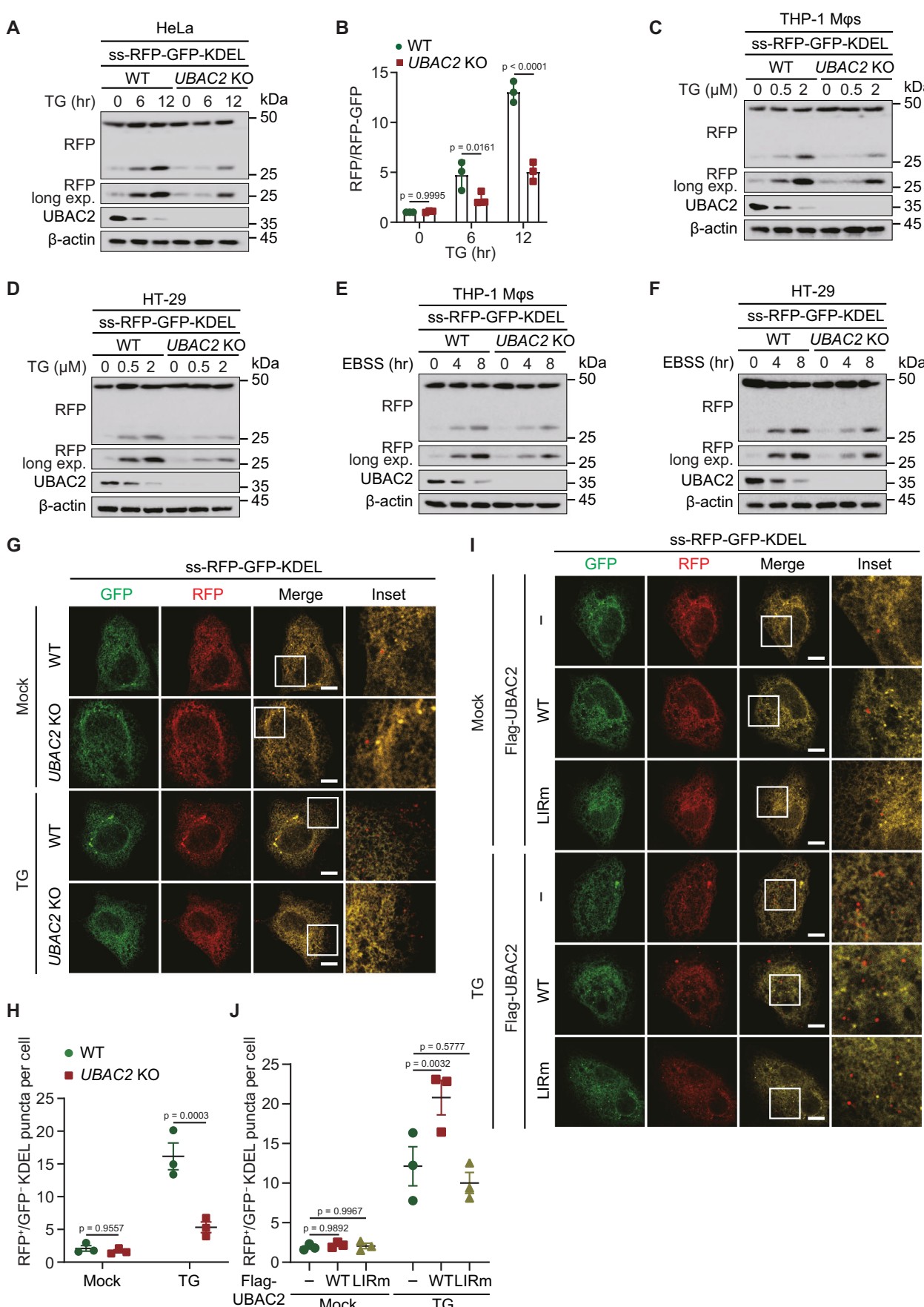

◀ **Figure EV2.  UBAC2 functions as an ER-phagy receptor.**

(**A**) HeLa cells stably expressing the ER-phagy reporter transfected with plasmid expressing Flag-UBAC2 were cultured in the presence of doxycycline for 24 h to induce the reporter. Lysates of the cells with TG (1 μM) for indicated time points were harvested for immunoblot analysis. (**B**) The band intensities of RFP and RFP-GFP in the similar samples as (**A**) from three independent experiments performed in duplicate were quantified and the ratios of RFP/RFP-GFP were shown. (**C, D**) WT or *UBAC2* knockout (KO) THP-1 Mφs (**C**) or HT-29 cells (**D**) stably expressing the ER-phagy reporter were cultured in the presence of doxycycline (Dox) (200 ng/mL) for 24 h to induce the reporter. Lysates of the cells treated with TG (1 μM) for indicated dosages for 8 h were harvested for immunoblot analysis. (**E, F**) WT or *UBAC2* KO Mφs (**E**) or HT-29 cells (**F**) stably expressing the ER-phagy reporter were cultured in the presence of Dox (200 ng/mL) for 24 h to induce the reporter. Lysates of the cells cultured in EBSS for indicated time points were harvested for immunoblot analysis. (**G**) WT or *UBAC2* KO HeLa cells stably expressing the ER-phagy reporter were treated with Dox (200 ng/mL) for 24 h. The cells were treated with or without TG (1 μM) for 8 h and were observed by fluorescence microscopy. Scale bar, 20 μm. (**H**) Quantitative analysis of the similar samples as (**G**) from three biologically independent experiments (20 cells scored per condition per experiment). (**I**) HeLa cells stably expressing the ER-phagy reporter were transfected with plasmids encoding WT or LIRm UBAC2 and treated with Dox (200 ng/mL) for 24 h. The cells were treated with or without TG (1 μM) for 8 h and observed by fluorescence microscopy. Scale bar, 20 μm. (**J**) Quantitative analysis of the similar samples as (**I**) from three biologically independent experiments (20 cells scored per condition per experiment). Data information: For (**A, C–G, I**), one representative experiment out of three was shown. In (**B, H, J**), data are presented as the mean ± SEM of three independent biological experiments. The statistical significance of the difference was analyzed by unpaired two-tailed Student's *t* test, and the *P* values were shown. Source data are available online for this figure.

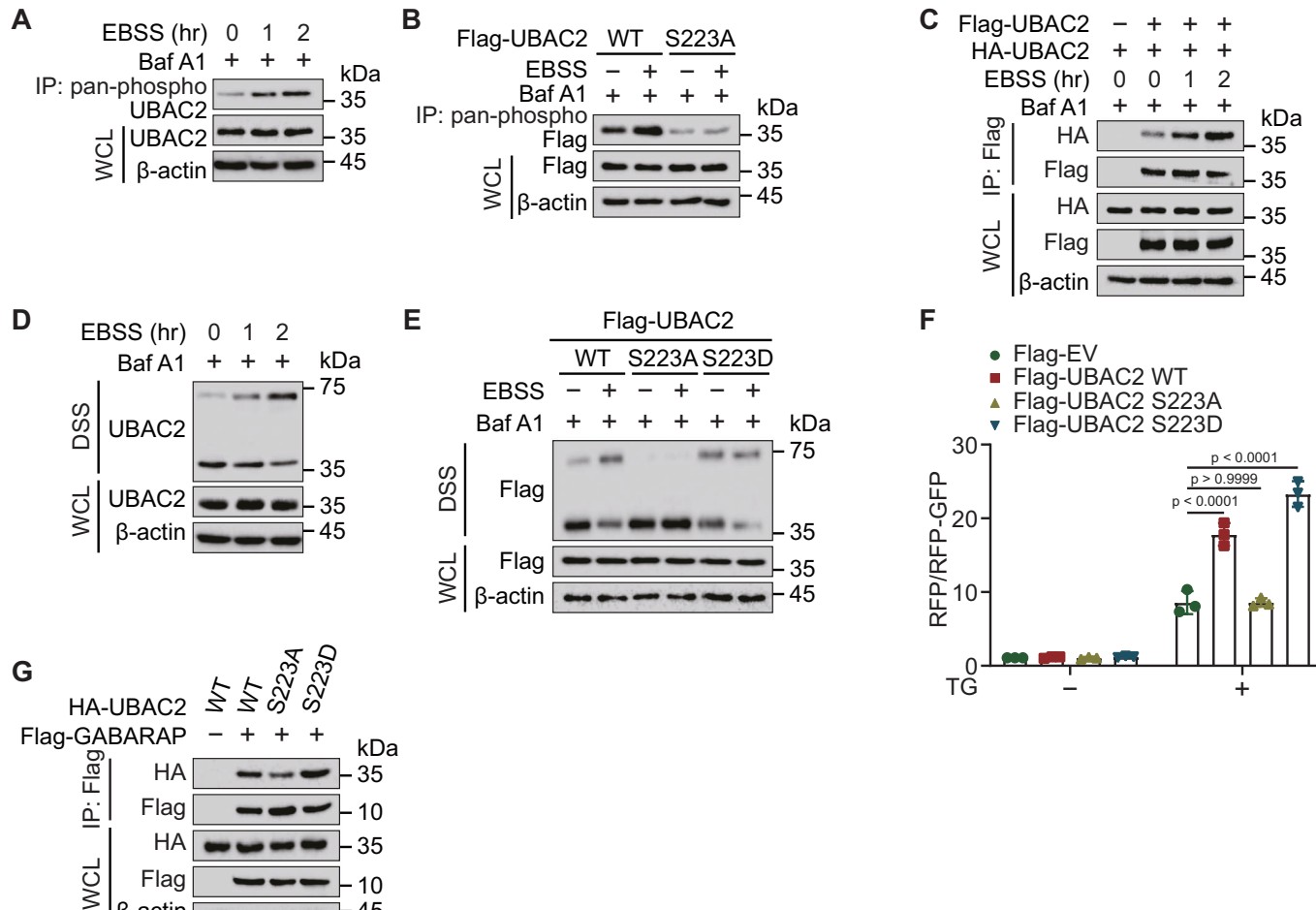

**Figure EV3.  S223 is critical for the phosphorylation and function of UBAC2.**

(A) Extracts of HeLa cells cultured with EBSS for indicated time points were immunoprecipitated using phosphoserine/threonine/tyrosine polyclonal antibody and immunoblotted with UBAC2 antibody. (B) 293T cells transfected with vector of Flag-UBAC2 or its S223A mutant were cultured within EBSS in the presence of Baf A1 (0.2 μM). The lysates were subjected to coimmunoprecipitation and immunoblot analysis. (C) Lysates of 293T cells transfected with plasmids for Flag-UBAC2 and HA-UBAC2, were subjected to immunoprecipitation with anti-Flag and immunoblot analysis with anti-HA. (D) Extracts of HeLa cells cultured with EBSS in the presence of Baf A1 (0.2 μM) were treated with 2 mM disuccinimidyl suberate (DSS) cross-linker and analyzed by immunoblotting. (E) 293T cells transfected with plasmids encoding WT UBAC2 or its indicated mutants were cultured in EBSS for 3 h with the existence of Baf A1 (0.2 μM). The lysates were treated with 2 mM disuccinimidyl suberate (DSS) cross-linker and analyzed by immunoblotting. (F) The band intensities of RFP and RFP-GFP in the similar samples as Fig. 4N from three independent experiments performed in duplicate were quantified and the ratios of RFP/RFP-GFP were shown. (G) Coimmunoprecipitation and immunoblot analysis of lysates from 293T cells transfected with vector of Flag-GABARAP and HA-UBAC2 as well as its indicated mutants. Data information: For (A–E, G), one representative experiment out of three was shown. In (F), data are presented as the mean ± SEM of three independent biological experiments. The statistical significance of the difference was analyzed by unpaired two-tailed Student's t test, and the P values were shown. Source data are available online for this figure.

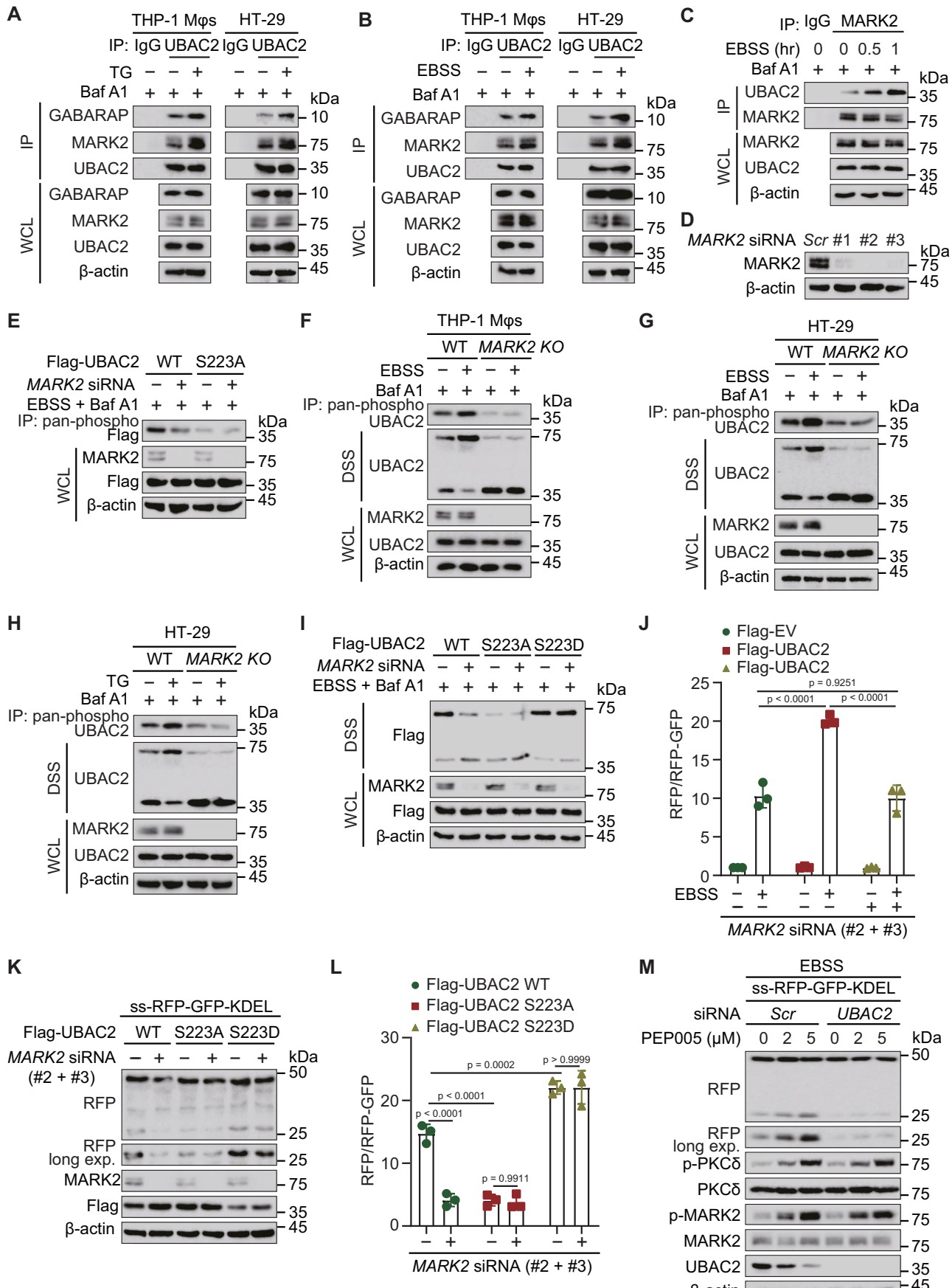

◀

**Figure EV4. MARK2 is needed for UBAC2-mediated ER-phagy.**

(A) Lysates of THP-1 Mφs (left) or HT-29 cells (right) treated with TG (1 μM) for 1 h in the presence of Baf A1 (0.2 μM) were subjected to immunoprecipitation and immunoblot analysis. (B) Lysates of THP-1 Mφs (left) or HT-29 cells (right) cultured in EBSS for 3 h together with Baf A1 (0.2 μM) treatment were subjected to immunoprecipitation and immunoblot analysis. (C) Lysates of HeLa cells treated with EBSS for indicated time points in the presence of Baf A1 (0.2 μM) were subjected to immunoprecipitation and immunoblot analysis. (D) Immunoblot analysis of the knockdown efficiency of MARK2 by *MARK2*-specific siRNAs in HeLa cells. (E) HeLa cells were transfected with scramble or *MARK2*-specific siRNA and treated with EBSS for 3 h in the presence of Baf A1 (0.2 μM). The lysates were subjected to immunoprecipitation and immunoblot analysis. (F) WT or *MARK2* KO THP-1 Mφs were cultured in EBSS for 3 h with the existence of Baf A1 (0.2 μM). The lysates were treated with 2 mM disuccinimidyl suberate (DSS) cross-linker or immunoprecipitated using phosphoserine/threonine/tyrosine polyclonal antibody, and detected by immunoblot. (G, H) WT or *MARK2* KO HT-29 cells were treated with TG (1 μM) for 1 h (G) and cultured in EBSS for 3 h (H) in the presence of Baf A1 (0.2 μM). The lysates were treated with 2 mM disuccinimidyl suberate (DSS) cross-linker or immunoprecipitated using phosphoserine/threonine/tyrosine polyclonal antibody, and detected by immunoblot. (I) 293T cells were transfected with scramble or *MARK2*-specific siRNA and then transfected with vector of WT or S223A UBAC2. After incubation the cells with EBSS for 3 h in the presence of Baf A1 (0.2 μM), the lysates were harvested for immunoprecipitation and immunoblot analysis. (J) The band intensities of RFP and RFP-GFP in the similar samples as Fig. 5K from three independent experiments performed in duplicate were quantified and the ratios of RFP/RFP-GFP were shown. (K) HeLa cells stably expressing the ER-phagy reporter were transfected with scramble or *MARK2*-specific siRNA alongside with WT or mutated UBAC2 plasmids. The cells were then cultured in the presence of doxycycline for 24 h to induce the reporter and the protein was harvested for immunoblot analysis. (L) The band intensities of RFP and RFP-GFP in the similar samples as (K) from three independent experiments performed in duplicate were quantified and the ratios of RFP/RFP-GFP were shown. (M) HeLa cells stably expressing the ER-phagy reporter were transfected with scramble or *UBAC2*-specific siRNA and then treated with PEP005 with indicated dosages for 1 h. The cells were cultured in EBSS for 3 h and the lysates were harvested for immunoblot analysis. Data information: For (A–I, K, M), one representative experiment out of three was shown. In (J, L), data are presented as the mean ± SEM of three independent biological experiments. The statistical significance of the difference was analyzed by unpaired two-tailed Student's *t* test, and the *P* values were shown. Source data are available online for this figure.

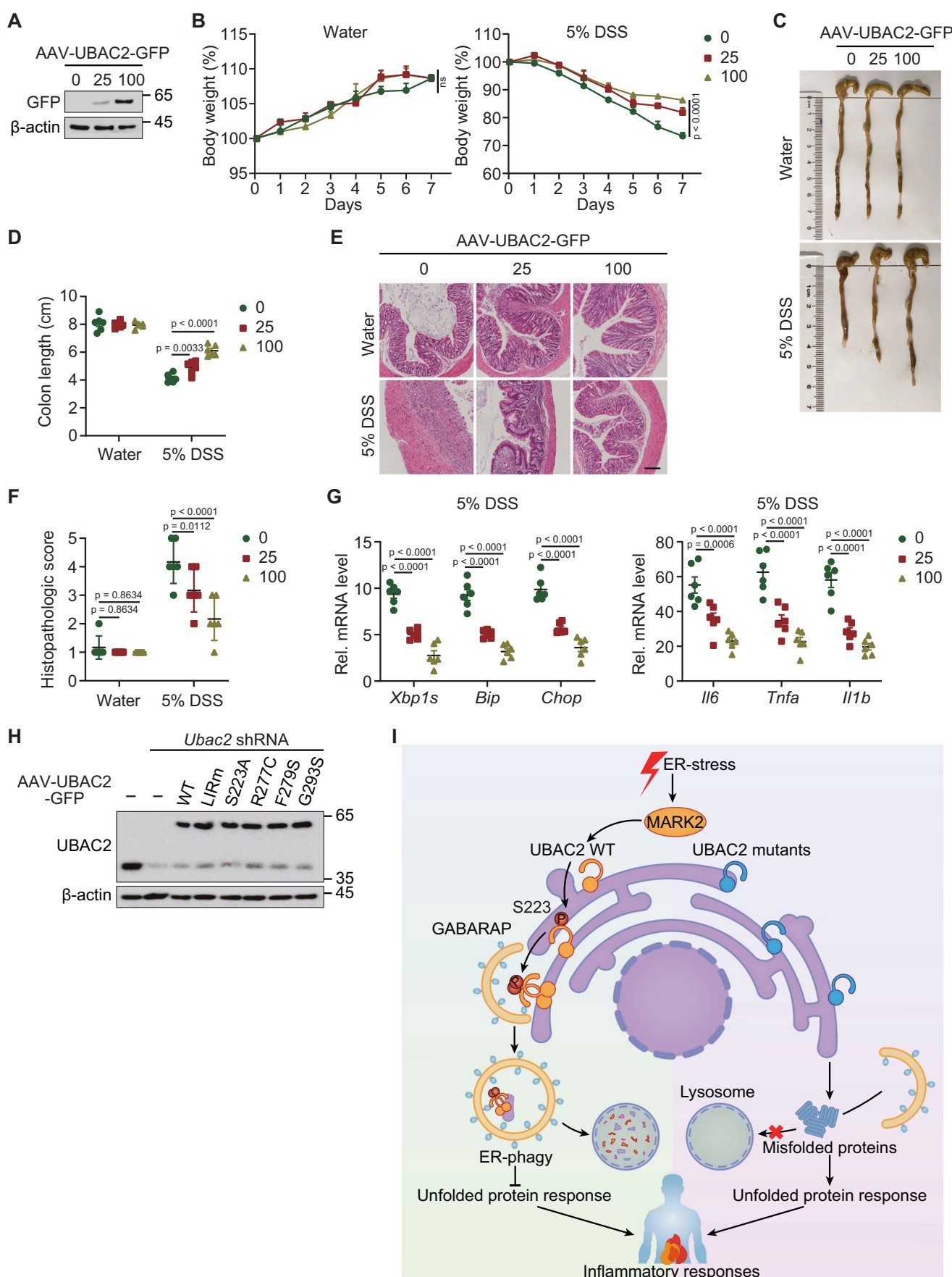

◀

**Figure EV5.   UBAC2-mediated ER-phagy inhibits DSS-induced colitis in mice.**

(A) Representative western blot image of GFP in intestine tissues of DSS-induced mice with increasing dosage of AAV-UBAC2-GFP treatment ($n = 6$ independent biological mice per group). (B) Changes in body weight (percentage of original body weight) over time (days) in water- or DSS-treated mice expressing UBAC2 with indicated dose ($n = 6$ independent biological mice per group). (C) Representative macroscopic features of colons from water- or DSS-treated mice expressing UBAC2 with indicated dose. (D) Colon length of water- or DSS-treated mice expressing UBAC2 with indicated dose ($n = 6$ independent biological mice per group). (E, F) Representative images (E) of H&E staining of colon sections and the histopathologic scores (F) from water- or DSS-treated mice expressing UBAC2 with indicated dose. Scale bar, 100 μm. Each experiment was repeated independently three times, and representative results are shown. (G) qPCR analysis of ER-stress inducible transcripts (left) and inflammatory genes (right) of intestine homogenates from DSS mice expressing UBAC2 with indicated dose. (H) Representative western blot image of UBAC2 in intestine tissues of DSS treated AAV-delivered shRNA *Ubac2* knockdown mice expressing WT and mutated UBAC2 ($n = 6$ independent biological mice per group). (I) A proposed working model to illustrate the regulation of UBAC2-directed ER-phagy in suppressing the inflammatory responses. Data information: In (B), data are presented as mean ± SEM ($n = 6$ independent biological mice per group). In (D, F, G), data are presented as mean ± SD ($n = 6$ independent biological mice per group). The statistical significance of the difference was analyzed by two-way ANOVA with Bonferroni's multiple comparisons test (B) or unpaired two-tailed Student's *t* test (D, F, G), and the *P* values were shown. Source data are available online for this figure.

