## [Peer Review File · The EMBO Journal]

ER-phagy restrains inflammatory responses through its receptor UBAC2

Xing He, Haowei He, Zitong Hou, Zheyu Wang, Qinglin Shi, Tao Zhou, Yaoxing Wu, Yunfei Qin, Jun Wang, Zhe Cai, Jun Cui, and Shouheng Jin

Corresponding author(s): Shouheng Jin (jinshh3@mail.sysu.edu.cn)

Review Timeline:

Submission Date:	5th Jan 24
Editorial Decision:	2nd Feb 24
Revision Received:	27th May 24
Editorial Decision:	7th Jul 24
Revision Received:	11th Jul 24
Accepted:	22nd Aug 24

Editor: William Teale

Transaction Report:

Dear Dr. Jin,

Thank you again for the submission of your manuscript entitled "ER-phagy restrains inflammatory responses through receptor UBAC2" and for your patience during the review process. We have now received the reports from the referees, which I copy below.

As you can see from their comments, the referees found your manuscript clear and potentially important. However, all referees pointed out a need for UBAC2 function to be integrated more widely into ER-phagy receptor function in general. This point will require your attention before your manuscript can be published in The EMBO Journal.

Having said this, based on the overall interest expressed in the reports, I would like to invite you to address the comments of all referees in a revised version of the manuscript. I should add that it is The EMBO Journal policy to allow only a single major round of revision and that it is therefore important to resolve the main concerns at this stage. I believe the concerns of the referees are reasonable and addressable; I think it would be beneficial for us to talk over Zoom in the near future (but maybe after your new-year celebrations) about your plans for the manuscript's revision. but please contact me if you have any questions, need further input on the referee comments or if you anticipate any problems in addressing any of their points. Please, follow the instructions below when preparing your manuscript for resubmission.

I would also like to point out that as a matter of policy, competing manuscripts published during this period will not be taken into consideration in our assessment of the novelty presented by your study ("scooping" protection). We have extended this 'scooping protection policy' beyond the usual 3 month revision timeline to cover the period required for a full revision to address the essential experimental issues. Please contact me if you see a paper with related content published elsewhere to discuss the appropriate course of action.

Again, please contact me at any time during revision if you need any help or have further questions.

Thank you very much again for the opportunity to consider your work for publication. I look forward to your revision.

Best regards,

William Teale

William Teale, Ph.D.
Editor
The EMBO Journal

When submitting your revised manuscript, please carefully review the instructions below and include the following items:

- 1) a .docx formatted version of the manuscript text (including legends for main figures, EV figures and tables). Please make sure that the changes are highlighted to be clearly visible.
- 2) individual production quality figure files as .eps, .tif, .jpg (one file per figure).
- 3) a .docx formatted letter INCLUDING the reviewers' reports and your detailed point-by-point response to their comments. As part of the EMBO Press transparent editorial process, the point-by-point response is part of the Review Process File (RPF), which will be published alongside your paper.
- 4) a complete author checklist, which you can download from our author guidelines ([https://wol-prod-cdn.literatumonline.com/pb-assets/embo-site/Author Checklist%20-%20EMBO%20J-1561436015657.xlsx](https://wol-prod-cdn.literatumonline.com/pb-assets/embo-site/Author%20Checklist%20-%20EMBO%20J-1561436015657.xlsx)). Please insert information in the checklist that is also reflected in the manuscript. The completed author checklist will also be part of the RPF.
- 5) Please note that all corresponding authors are required to supply an ORCID ID for their name upon submission of a revised manuscript.
- 6) We require a 'Data Availability' section after the Materials and Methods. Before submitting your revision, primary datasets produced in this study need to be deposited in an appropriate public database, and the accession numbers and database listed

under 'Data Availability'. Please remember to provide a reviewer password if the datasets are not yet public (see <https://www.embopress.org/page/journal/14602075/authorguide#datadeposition>). If no data deposition in external databases is needed for this paper, please then state in this section: This study includes no data deposited in external repositories. Note that the Data Availability Section is restricted to new primary data that are part of this study.

Note - All links should resolve to a page where the data can be accessed.

8) For data quantification: please specify the name of the statistical test used to generate error bars and P values, the number (n) of independent experiments (specify technical or biological replicates) underlying each data point and the test used to calculate p-values in each figure legend. The figure legends should contain a basic description of n, P and the test applied. Graphs must include a description of the bars and the error bars (s.d., s.e.m.).

9) We would also encourage you to include the source data for figure panels that show essential data. Numerical data can be provided as individual .xls or .csv files (including a tab describing the data). For 'blots' or microscopy, uncropped images should be submitted (using a zip archive or a single pdf per main figure if multiple images need to be supplied for one panel). Additional information on source data and instruction on how to label the files are available at .

10) We replaced Supplementary Information with Expanded View (EV) Figures and Tables that are collapsible/expandable online (see examples in <https://www.embopress.org/doi/10.15252/embj.201695874>). A maximum of 5 EV Figures can be typeset. EV Figures should be cited as 'Figure EV1, Figure EV2" etc. in the text and their respective legends should be included in the main text after the legends of regular figures.

12) Our journal encourages inclusion of *data citations in the reference list* to directly cite datasets that were re-used and obtained from public databases. Data citations in the article text are distinct from normal bibliographical citations and should directly link to the database records from which the data can be accessed. In the main text, data citations are formatted as follows: "Data ref: Smith et al, 2001" or "Data ref: NCBI Sequence Read Archive PRJNA342805, 2017". In the Reference list, data citations must be labeled with "[DATASET]". A data reference must provide the database name, accession number/identifiers and a resolvable link to the landing page from which the data can be accessed at the end of the reference. Further instructions are available at .

Further instructions for preparing your revised manuscript:

We realize that it is difficult to revise to a specific deadline. In the interest of protecting the conceptual advance provided by the work, we recommend a revision within 3 months (2nd May 2024). Please discuss the revision progress ahead of this time with the editor if you require more time to complete the revisions. Use the link below to submit your revision:

Referee #1:

In this manuscript, He et al., claimed to identify a new ER-phagy receptor that could restrain inflammatory response. They showed that UBAC2 interacts with the LIR domain of GABARAP. They identified MARK2 as a kinase for UBAC2 important for ER-phagy. Also, they showed the significance of the study in mice models of IBD. Overall, the experiments are clean. However, most of the experiments are performed overexpression system in HEK293T cells. Also, almost all endogenous data is presented in one cell line, HeLa cells, and 1/2 siRNA. This reviewer considers this as a major drawback of the manuscript and strongly suggests using at least 2 more cell lines in some of the MAJOR endogenous experiments. The choice of experiments remains with the authors. The use of some other method of KD (stable) or KO (CRISPR) method should also be considered. This will improve the trust in the versatility and importance of the discovery. In the absence, this reviewer may not endorse it for publication.

Major concerns

Figure 1

Panel J-O; The increase in co-localization appears to be because of an increase in WIPI2, ATG16L, or LAMP, rather than UBAC2 puncta formation.

Also, please provide pictures with at least 3-4 cells in each field with similar results in Supplementary Figures (everywhere where immunofluorescence data is shown). Single-cell images don't provide confidence for quantitative analysis.

Figure 2

Panel B, C- Why does UBAC2 don't accumulate with BafA1 if it's targeted by autophagy?

Panel G- It's interesting to see even no background interaction with LIRm.

Figure 3

Please perform UBAC2 co-localization with ER markers

If UBAC2 is an ER-phagy receptor then KD/KO of UBAC2 should increase total ER content. Please use western blot experiments with ER markers (or total ER fractions). %ER area increase in data in panel F, I is not convincing from the images given so please perform Western blots.

Figure 5

Please perform interaction assays and invitro phosphorylation assays with MARK2 and UBAC2 to clearly demonstrate that MARK2 is the kinase.

Panel K. Data not convincing. Please perform western blotting with ER markers to show an increase.

Figure 6.

Panel C, D Please perform western blotting with ER stress markers to show increase/decrease.

Panel E, F Please perform western blotting with inflammatory markers to show an increase/decrease.

As the authors in the introduction pointed out about interferon response induction during UPR. Please check interferon response with western blots with ISGs (Such as ISG15 and MX1).

Referee #2:

He et al. present a study in which they analyzed the function of the ubiquitin-associated domain-containing protein 2 (UBAC2) in the degradation of parts of the endoplasmic reticulum (ER) by autophagy (ERphagy).

Using an unbiased proximity ligation approach to identify proteins involved in ERphagy, He et al. identified UBAC2 as a potential hit. They found that the degradation of UBAC2 depends on autophagy, that UBAC2 is recruited to the ER upon ER stress and that UBAC2 contains a LIR motif that binds preferentially to GABARAP. Using RFP-GFP-KDEL as a marker for ERphagy, the authors demonstrated that UBAC2 functions as an autophagy receptor and mediates ERphagy during ER stress. They also found that the activity of UBAC2 is regulated by the MARK2 kinase that phosphorylates S223 of UBAC2. ER phagy was inhibited if this serine residue was mutated to alanine.

Impaired quality control of the ER induces inflammation and mutations in UBAC2 are associated with inflammatory diseases. The authors investigated whether the function of UBAC2 in ER phagy ameliorates inflammation of the intestine upon treatment of mice with Dextran sulfate sodium (DSS), which is a model for inflammatory diseases. They found that UBAC2, but not mutants associated with inflammatory diseases, dampened inflammation upon DSS treatment, reduced the production of inflammatory cytokines and reduced infiltration of inflammatory cells.

In general, the study is very well designed and the experiments presented in a clear and logic manner. The data are overall convincing. What is missing in the study is the characterization of the relationship of UBAC2 with the other known ERphagy receptors such as Fam134B, RTN3L, ATL3, Sec62, CCPG1, TEX264 and C53.

This study will merit publication in the EMBO journal if the authors addressed concerns that are listed below.

Comments:

The manuscript would greatly benefit from additional experiments to characterize how other ER phagy receptors crosstalk with UBAC2. Colocalization of UBAC2 with Fam134B, RTN3L and ALT3 should be provided and ER phagy flux should be analyzed in cells after siRNA mediated KD of combinations of UBAC2 with other receptors.

Figure 1: the contrast settings of micrographs in panel M and N are different between MOCK (high contrast) and EBSS (low contrast). The authors should apply identical contrast settings to MOCK and EBSS images, if at all required. Please also provide information where the original, unmodified raw images can be found (Supplement or data repertoire).

Quantifications of interactions in co-IP experiments should be provided (e.g. Fig. 1F, H, I; Fig. 2A, B, C, and related EV Figures). From the presented data in Fig. 2A and its corresponding legend, it appears that binding of UBAC2 to ATG8 proteins was only analyzed under fed conditions. However, given the scope of the manuscript and the major conclusions, similar experiments under starvation conditions and upon treatment with Tg need to be provided. This is important since other ATG8 protein might interact in particular during autophagy induced conditions with UBAC2. If there is a considerable binding to other ATG8s in starved or Tg treated cells, the LIR mutant needs to be tested for its binding to these ATG8s as well.

ERphagy was analyzed using the RFP-GFP-KDEL construct and quantifying the number of red positive and green negative puncta. Why was this analysis done in UBAC2KO cells for western blot analysis, but in siRNA knockdown cells for confocal microscopy (Fig 3C and D)? The experiment in Fig 3D should be repeated with KO cells.

Moreover, in the western blot of RFP cleavage assays in Fig. 4N a strong double band is detected in Tg treated samples for the control, but for nontreated samples for S223A and S223D, while no such double band was detected in Fig. 3B. What is the reason for this double band and why does it occur only in some samples?

Also, ERphagy was induced by starvation and not by ER stress using Tg in Fig. 3C and experiments in Fig. 3 D-I were performed in fed cells. The experiments should be repeated in Tg treated cells.

Information on the loaded total protein is missing for all co-IP experiments. Was the input diluted and was the protein after co-IP resuspended in a similar volume of buffer (compared to the volume of lysed cells)? What is the difference between WCL and input?

A phosphorylation site in UBAC2 was identified (S223), but a significant residual phosphorylation of the protein was detected in a phosphorylation deficient mutant (S223A). What are these other phosphorylation sites, have they been detected by mass-spec? If the band in the IP experiment with the S223A mutant reflects nonspecific binding, the band should also be present after dephosphorylation with a nonselective phosphatase. The samples in Fig 4C (wt and mut) should be treated with a phosphatase and co-IP with the pan-phospho antibody should be repeated. This control is also important since the experiment was performed with a pan-phospho antibody and not a selective one.

UBAC2 was found to be phosphorylated by MARK2. In Fig. 5D phosphorylation of WT and S223A variants of UBAC2 were tested in control and MARK2 KD conditions. The experiment should be repeated including the phosphomimic S223D variant. The finding that UBAC2 expression in mice protects against inflammation in a model of DSS induced colitis are interesting, but the section appears to be disconnected from the mechanistic characterization of UBAC2. The authors should expand on the rationale for this experiment and analyze the UBAC2 mutants that are associated with human diseases using the RFP-GFP-KDEL construct.

Minor points:

On page 9 line 13 it should read ... when S223 was mutated to alanine.

Amount of puncta should be replaced by number of puncta in the text.
Why is no input (WCL) shown for the IgG control in Fig. 5B?

Referee #3:

The endoplasmic reticulum (ER), as the largest continuous membranous system within cells, plays a pivotal role in essential cellular processes such as protein synthesis, lipid metabolism, immune response, and organelle biogenesis. ER-phagy, the autophagy of the endoplasmic reticulum, is essential for maintaining ER homeostasis, and various ER-phagy receptors have been identified in previous studies. A significant scientific question in this field is whether there are additional ER-phagy receptors that respond to different ER stresses.

In this manuscript, He and colleagues propose a novel ER-phagy receptor, UBAC2. Under inflammatory stimuli, UBAC2 undergoes phosphorylation modifications, leading to dimerization and activation of ER-phagy. In a dextran sodium sulfate (DSS)-induced inflammatory model, this biological process alleviates inflammation. Overall, the data in this manuscript are of high quality, the figures are pleasing, and the experimental phenotypes are evident. However, significant flaws exist in some experimental designs and logical coherence, making it potentially unsuitable for publication in the *Embo Journal*.

Major comments

1: CCPG1 was previously considered one of the ER-phagy receptors in earlier studies. Authors observed a significant increase in its protein levels under conditions of TG treatment, using this as a crucial indicator in their screening work. What is the underlying mechanism for this? Authors maybe should present all previously recognized ER-phagy receptors and other clearly evident ER-phagy substrates in this context.

2: Authors demonstrates the co-localization of UBAC2 with proteins such as ATG16L, LAMP1, GABARAP, etc., and suggests that the starvation treatment results in an increased co-localization, possibly due to a significant increase in the abundance of the associated proteins.

3: As a calcium ion inhibitor mediated by SERCAs, what is the direct mechanism of TG on UBAC2? Does UBAC2 directly bind to TG molecules, or does it depend on the functionality of other ER-phagy receptors?

4: Throughout the entire manuscript, authors lack a comparison of UBAC2 with all currently identified ER-phagy receptors. Does UBAC2 function independently of other ER-phagy receptors? What is the proportion of UBAC2's involvement in the entire ER-phagy process?

5: Authors attempt to elucidate in the manuscript that the dimerization of UBAC2 is a functional form for its activity. What significance does this dimerization hold, and in essence, why does UBAC2 require dimerization to function? Is dimerization dependent on its transmembrane domain? Can the analysis of dimerization forms provide additional truncated data? Can it offer in vitro data or data from Native gel experiments?

6: The phosphorylation sites identified by authors are distant from the ER membrane. How does phosphorylation by MARK2 affect the dimerization of UBAC2? What is the upstream signal of this process? Does the activation of MARK2 lead to an increase in ER-phagy levels?

7: In Figure 6, it would be more suitable for authors to use Tunicamycin rather than Thapsigargin to validate the proposed model.

8: In Figure 7, in the colitis model experiments, it appears that authors need to supplement the differences between UBAC2 KO mice and WT mice, and to perform rescue experiments with various mutants in UBAC2 KO. Because UBAC2 exists in a dimeric form, and the authors did not resolve the details of this form, the endogenous UBAC2 in WT mice and the expression of various mutant forms may introduce interference.

Minor comments

1: In the confocal experiments, the imaging of UBAC is blurry, and HeLa cell morphology is poor.

2: Figure 2L, the confocal image is misplaced.

3: For all in vitro experiments, please supplement the target protein's purified samples with Coomassie Brilliant Blue staining or other data confirming protein purity.

4: For the WB experiments under TG treatment, please include ER-phagy substrates as positive controls.

5: Authors should statistically present the RFP-GFP confocal experiment as a ratio rather than simply counting the number of RFP points.

6: Figure 4G and I, under the same treatment conditions, there is a significant disparity in the number of UBAC2-positive puncta in the Mock group. This suggests experimental instability.

7: To substantiate MARK2 as the kinase for UBAC2, additional data such as in vitro phosphorylation need to be provided.

8: Figure 5k, how can ER-tracker be used to quantify the size of the endoplasmic reticulum (ER)? Do the authors propose alternative and potentially more effective methods for characterizing ER morphology?

9 : Given authors' attempt to establish UBAC2 as a novel ER-phagy receptor and their belief that the punctate UBAC2 structures observed in co-localization experiments indicate autophagic functionality, it is necessary for the authors to supplement staining with LAMP1 and LC3 in all GFP-UBAC2 punctate structure experiments.

10 : Some portions of DSS were incorrectly labeled as SDS.

Dear Referees,

Thank you very much for providing us a valuable opportunity to revise our paper entitled “*ER-phagy restrains inflammatory responses through receptor UBAC2*”. We have performed additional experiments and provided new data in accordance with the Referees’ suggestions. These results strengthen our overall conclusions and profoundly improve the quality of the manuscript. A point-by-point response to the Referees’ concerns is included below. For the convenience, we have numbered all the suggestions and pasted all the figures in response to each question in this letter. We have also highlighted our modification with yellow in our text.

We hope that the revised manuscript meets the requirements for publication in “*EMBO Journal*”, and we look forward to hearing from you.

Sincerely,

Shouheng Jin, Ph.D.

Associate Professor of Department of Biochemistry

School of Life Sciences

Sun Yat-sen University

(The corresponding author)

Response to the comments of Reviewer #1

1. In this manuscript, He et al., claimed to identify a new ER-phagy receptor that could restrain inflammatory response. They showed that UBAC2 interacts with the LIR domain of GABARAP. They identified MARK2 as a kinase for UBAC2 important for ER-phagy. Also, they showed the significance of the study in mice models of IBD. Overall, the experiments are clean. However, most of the experiments are performed overexpression system in HEK293T cells. Also, almost all endogenous data is presented in one cell line, HeLa cells, and 1/2 siRNA. This reviewer considers this as a major drawback of the manuscript and strongly suggests using at least 2 more cell lines in some of the MAJOR endogenous experiments. The choice of experiments remains with the authors. The use of some other method of KD (stable) or KO (CRISPR) method should also be considered. This will improve the trust in the versatility and importance of the discovery. In the absence, this reviewer may not endorse it for publication.

Response: We thank the reviewer for this comment to improve our manuscript and have performed a series of experiments according to the reviewer's suggestion. We have repeated almost all our key findings in THP-1 (a human monocytic cell line which can be differentiated into macrophage) and HT-29 (a human colorectal adenocarcinoma cell line) cells. We observed that the thapsigargin (TG)-induced UBAC2 degradation was blocked by bafilomycin A1 (Baf A1) treatment both in THP-1-derived macrophages (THP-1 Mφs) and HT-29 cells (**New Figures 1A and 1B, also see Appendix Fig S1B,C**). UBAC2 could undergo autophagic degradation under Earle's balanced salt solution (EBSS)-cultured starvation conditions in THP-1 Mφs and HT-29 cells (**New Figures 1C and 1D, also see Appendix Fig S1D,E**). In UBAC2 KO cells, the TG or EBSS induced ER-phagy flux was greatly diminished (**New Figures 1E–1H, also see Fig. EV2C–F**). Additionally, UBAC2-GABARAP association and UBAC2-MARK2 association were increased by TG or EBSS treatment in THP-1 Mφs and HT-29 cells (**New Figures 1I–1L, also see Fig. EV4A,B**). The increased phosphorylation and dimerization of UBAC2 were abrogated in MARK2 KO THP-1 Mφs and HT-29 cells (**New Figures 1M–1P, also**

see **Figs. 5F and EV4F–H**). Taken together, these new results further indicate that UBAC2 is an ER-phagy receptor, and can be phosphorylated by MARK2 to undergo dimerization to facilitate ER-phagy flux.

2. Figure 1 Panel J-O; The increase in co-localization appears to be because of an increase in WIPI2, ATG16L, or LAMP, rather than UBAC2 puncta formation. Also, please provide pictures with at least 3-4 cells in each field with similar results in Supplementary Figures (everywhere where immunofluorescence data is shown). Single-cell images don't provide confidence for quantitative analysis.

Response: We thank the reviewer for pointing out this question. We have conducted confocal microscopic analysis and obtained better images to show the co-localization between UBAC2 and components of autophagy machinery. Due to the autophagic degradation of UBAC2 upon autophagy activation conditions, we repeated our experiments with Baf A1 treatment to block the autolysosomal degradation. Our high-resolution images revealed that autophagy activation increased the formation of UBAC2 puncta. Moreover, the co-localization between UBAC2 and components of autophagy machinery was significantly enhanced upon autophagy activation (**New Figures 2A–2F, also see Figs. 1J–O**). Coimmunoprecipitation (Co-IP) and immunoblot analysis showed that the interaction between UBAC2 with WIPI2, ATG16L or LAMP1 was increased under the starvation-induced autophagy conditions (**New Figure 2G, also see Appendix Fig S1H**). We have shown about 3 cells in each field for our confocal microscopic analysis as suggested by the reviewer (**Fig. 2D, Fig. 2H, Fig. 2L, Fig. 3D, Fig. 3G, Fig. 4G, Fig. 4I, and Fig. 5H**). Moreover, we have quantified the confocal microscopy images from three biologically independent experiments (20 cells scored per condition per experiment). The numbers of cells for statistical analysis meets the standard for co-localization assay as previously described in several different research fields (Li *et al*, 2022; Newman *et al*, 2024; Wang *et al*, 2017).

3. Figure 2 Panel B, C- Why does UBAC2 don't accumulate with BafA1 if it's targeted by autophagy? Panel G- It's interesting to see even no background interaction with LIRm.

Response: To compare the interaction strength between UBAC2 and GABARAP, we have added Baf A1 to inhibit the autolysosomal degradation of UBAC2 in these experimental settings. Briefly, cells cultured in Baf A1 for 3 hr were treated with thapsigargin (TG) for indicated time points. Baf A1 treatment resulted in the equality of protein abundances of UBAC2, thus makes the samples suitable for further immunoprecipitation and immunoblot analysis. Additionally, the association between GABARAP and UBAC2 almost totally disappeared when the LIR motif of UBAC2 was mutated. Our results indicate that the LIR localized at 275–278 amino acid of UBAC2 is essential for the UBAC2-GABARAP interaction.

4. Figure 3 Please perform UBAC2 co-localization with ER markers. If UBAC2 is an ER-phagy receptor then KD/KO of UBAC2 should increase total ER content. Please use western blot experiments with ER markers (or total ER fractions). %ER area increase in data in panel F, I is not convincing from the images given so please perform Western blots.

Response: We thank the reviewer for this kind suggestion. Firstly, we studied the co-localization of UBAC2 with receptor accessory protein (REEP5) and calnexin (CANX), both are ER markers. Confocal microscopic analysis revealed that UBAC2 could co-localize with REEP5 and CANX (**New Figure 3A**). Secondly, we detected the protein abundances of REEP5 and CLIMP-63 (cytoskeleton-linking ER membrane protein of 63 kDa, an ER shaping protein), finding that the decreased ER content (monitored by detecting the REEP5 and CLIMP-63 abundances) induced by starvation was largely abrogated in *UBAC2* KO cells (**New Figure 3B**, also see **Fig. 3F**). Furthermore, the reduced ER content by UBAC2 under starvation conditions almost disappeared when the LIR of UBAC2 was mutated (**New Figure 3C**, also see **Fig. 3I**). Our western blot results further demonstrate that UBAC2 promotes the process of ER-phagy in an LIR-mediated manner. We have replaced the quantification of ER content from fluorescence microscopy images into immunoblot analysis.

5. Figure 5 Please perform interaction assays and *in vitro* phosphorylation assays with MARK2 and UBAC2 to clearly demonstrate that MARK2 is the kinase. Panel K. Data not convincing. Please perform western blotting with ER markers to show an increase.

Response: We performed *in vitro* binding and phosphorylation assay according to the reviewer's suggestion. We observed that purified UBAC2 from *E. coli* could interact with purified MARK2 from 293T cells (**New Figure 4A, also see Fig. 5C**). Moreover, the incubation of UBAC2 with MARK2 increased the phosphorylation of WT UBAC2, but not the S223A mutated form of UBAC2 *in vitro* (**New Figure 4B, also see Fig. 5E**). Immunoblotting study further revealed that UBAC2 S223D mutant, but not S223A mutated form of UBAC2 had a stronger ability to reduce ER content (monitored by detecting the abundances of ER markers) (**New Figure 4C, also see Fig. 5L**). Collectively, our results indicate that MARK2 phosphorylates UBAC2 at S223, which is crucial for the role of UBAC2 in ER-phagy.

6. Figure 6 Panel C, D Please perform western blotting with ER stress markers to show increase/decrease. Panel E, F Please perform western blotting with inflammatory markers to show an increase/decrease. As the authors in the introduction pointed out about interferon response induction during UPR. Please check interferon response with western blots with ISGs (Such as ISG15 and MX1).

Response: We thank the reviewer for this constructive suggestion. We performed western blotting analysis and found that *UBAC2* KO increased the protein abundances of ER stress markers (BiP, CHOP and XBP1S) upon TG or tunicamycin (TM) treatment. We also detected the activation of NF- κ B signaling pathway, observing that *UBAC2* depletion enhanced the phosphorylation of p65 induced by TG or TM. Moreover, we found that *UBAC2* KO increased the protein levels of IFIT1 and MX1 (both are encoded by ISGs) (**New Figure 5A, also see Appendix Fig S2G**). Additionally, only WT UBAC2, but not LIR mutated UBAC2 reduced the protein abundances of ER stress markers, the phosphorylation of p65, alongside with the expression levels of ISG-encoded products (**New Figure 5B, also see Appendix Fig S2G**).

Response to the comments of Reviewer #2

1. He et al. present a study in which they analyzed the function of the ubiquitin-associated domain-containing protein 2 (UBAC2) in the degradation of parts of the endoplasmic reticulum (ER) by autophagy (ERphagy). Using an unbiased proximity ligation approach to identify proteins involved in ERphagy, He et al. identified UBAC2 as a potential hit. They found that the degradation of UBAC2 depends on autophagy, that UBAC2 is recruited to the ER upon ER stress and that UBAC2 contains a LIR motif that binds preferentially to GABARAP. Using RFP-GFP-KDEL as a marker for ERphagy, the authors demonstrated that UBAC2 functions as an autophagy receptor and mediates ERphagy during ER stress. They also found that the activity of UBAC2 is regulated by the MARK2 kinase that phosphorylates S223 of UBAC2. ER phagy was inhibited if this serine residue was mutated to alanine. Impaired quality control of the ER induces inflammation and mutations in UBAC2 are associated with inflammatory diseases. The authors investigated whether the function of UBAC2 in ER phagy ameliorates inflammation of the intestine upon treatment of mice with Dextran sulfate sodium (DSS), which is a model for inflammatory diseases. They found that UBAC2, but not mutants associated with inflammatory diseases, dampened inflammation upon DSS treatment, reduced the production of inflammatory cytokines and reduced infiltration of inflammatory cells. In general, the study is very well designed and the experiments presented in a clear and logic manner. The data are overall convincing. What is missing in the study is the characterization of the relationship of UBAC2 with the other known ERphagy receptors such as Fam134B, RTN3L, ATL3, Sec62, CCPG1, TEX264 and C53. This study will merit publication in the EMBO journal if the authors addressed concerns that are listed below.

Response: We sincerely thank the reviewer for this positive comment and have performed additional experiments to reinforce our conclusions as suggested by the reviewer.

2. *The manuscript would greatly benefit from additional experiments to characterize how other ER phagy receptors crosstalk with UBAC2. Colocalization of UBAC2 with Fam134B, RTN3L and ALT3 should be provided and ER phagy flux should be analyzed in cells after siRNA mediated KD of combinations of UBAC2 with other receptors.*

Response: To clarify the relationship of UBAC2 with the other known ER-phagy receptors, we performed Co-IP and immunoblot analysis and found that UBAC2 had only a weak interaction with FAM134B. However, the interaction between UBAC2 and FAM134B was not affected by starvation-induced autophagy activation (**New Figure 6A, also see Fig. 3K**). Confocal analysis also revealed that FAM134B could partially colocalize with UBAC2 (**New Figure 6B**). To further study whether FAM134B is essential for the ER turnover mediated by UBAC2, we knocked down the expression of *FAM134B* and observed that UBAC2-promoted ER-phagy was not affected in *FAM134B*-deficient cells (**New Figure 6C, also see Fig. 3L**). Our results reveal that the role of UBAC2 in promoting ER-phagy flux is relatively independent.

3. *Figure 1: the contrast settings of micrographs in panel M and N are different between MOCK (high contrast) and EBSS (low contrast). The authors should apply identical contrast settings to MOCK and EBSS images, if at all required. Please also provide information where the original, unmodified raw images can be found (Supplement or data repertoire).*

Response: To resolve the reviewer's concern, we have presented more cells in each field for our confocal microscopic analysis. Our high-resolution images showed that the contrast settings of micrographs in different panels were similar and the previous difference was caused by the compression during the image export. Our updated data further indicated that the co-localization between UBAC2 and components of autophagy machinery was significantly enhanced with autophagy activation. Also see Response to Comment #2 of Reviewer #1. We have also uploaded our raw confocal microscopic images in Source Data Figures.

4. *Quantifications of interactions in co-IP experiments should be provided (e.g. Fig. 1F, H, I; Fig. 2A, B, C, and related EV Figures).*

Response: We have added the quantification of our Co-IP experiments according to the reviewer's suggestion in **New Figure 7**.

5. *From the presented data in Fig. 2A and its corresponding legend, it appears that binding of UBAC2 to ATG8 proteins was only analyzed under fed conditions. However, given the scope of the manuscript and the major conclusions, similar experiments under starvation conditions and upon treatment with Tg need to be provided. This is important since other ATG8 protein might interact in particular during autophagy induced conditions with UBAC2. If there is a considerable binding to other ATG8s in starved or Tg treated cells, the LIR mutant needs to be tested for its binding to these ATG8s as well.*

Response: We have examined the interaction between UBAC2 and ATG8 family proteins under TG-induced ER stress or EBSS-induced starvation condition. Our results showed that UBAC2 most strongly interacted with GABARAP upon EBSS or TG treatment. The weak association between UBAC2 and other ATG8 proteins was not obviously influenced under TG-induced ER stress or EBSS-induced starvation condition comparing to Mock-treated groups (**New Figures 8A and 8B, also see Fig. EV1B,C**). Our results indicate that GABARAP is a major target for UBAC2-mediated ER-phagy and we have chosen it for our further mechanism study.

6. ERphagy was analyzed using the RFP-GFP-KDEL construct and quantifying the number of red positive and green negative puncta. Why was this analysis done in UBAC2KO cells for western blot analysis, but in siRNA knockdown cells for confocal microscopy (Fig 3C and D)? The experiment in Fig 3D should be repeated with KO cells.

Response: Since our independent experiments were performed at a long period of time, we have chosen both knockdown and knockout assay in this study. We have verified the ER-phagy flux in knocking out system as suggested by the reviewer. Our data revealed that *UBAC2* KO reduced the ER-phagy flux (**New Figures 9A–9C, also see Fig. 3D,E**).

8. Also, ERphagy was induced by starvation and not by ER stress using Tg in Fig. 3C and experiments in Fig. 3 D-I were performed in fed cells. The experiments should be repeated in Tg treated cells.

Response: We apologized that we did not label the results in our previous Fig. 3D–I clearly as these experiments were performed in HeLa cells with EBSS treatment. Since the basal level of UBAC2-mediated ER-phagy under fed condition was quite weak, we studied the molecular mechanism of UBAC2 in ER-phagy under EBSS or TG treatment. We detected the ER-phagy flux with TG treatment, and observed that *UBAC2* KO significantly reduced the production of RFP fragments after treated with TG (New Figure 11A, also see Fig. EV2A,B). We also added the TG-treated groups to repeat the previous Fig. 3D–I (New Figures 11C–11H, also see Fig. EV2G–J). Our results further indicate that UBAC2 is critical to mediate ER-phagy under both EBSS-induced starvation and TG-treated ER stress conditions.

9. Information on the loaded total protein is missing for all co-IP experiments. Was the input diluted and was the protein after co-IP resuspended in a similar volume of buffer (compared to the volume of lysed cells)? What is the difference between WCL and input?

Response: We apologize for causing this misunderstanding. For the samples of whole cell lysates in the Co-IP experiments to detect the interaction between proteins, the label of “WCL” and “Input” indicates the same things. We have modified our labeling consistently in our figures with WCL. We used 5% of the whole cell lysates for WCL analysis, and the other 95% for Co-IP assay.

10. A phosphorylation site in UBAC2 was identified (S223), but a significant residual phosphorylation of the protein was detected in a phosphorylation deficient mutant (S223A). What are these other phosphorylation sites, have they been detected by mass-spec? If the band in the IP experiment with the S223A mutant reflects nonspecific binding, the band should also be present after dephosphorylation with a nonselective phosphatase. The samples in Fig 4C (wt and mut) should be treated with a phosphatase and co-IP with the pan-phospho antibody should be repeated. This control is also important since the experiment was performed with a pan-phospho antibody and not a selective one.

Response: In our mass spectrometry analysis, we also found a phosphorylation site of UBAC2 at serine (S) 24. Since S24 localizes in the ER lumen, we have ignored this site in the regulation of ER-phagy. To study the possibility of this phosphorylation site in the modulation of UBAC2-mediated ER-phagy, we generated the S24A mutated form of UBAC2. We found that the increased phosphorylation of UBAC2 by TG treatment was also abrogated when the S223, but not S24 of UBAC2 was mutated (**New Figure 12A**). To further check whether the band detected by pan-phospho antibody is phosphorylated UBAC2, we repeated our previous Co-IP experiment in the existence of a phosphatase. We observed that the bands of Flag-tagged UBAC2 after immunoprecipitation with pan-phospho antibody were totally disappeared with a phosphatase incubation (**New Figure 12B**). All these results indicate that the S223 is important for the phosphorylation of UBAC2 in mediating ER-phagy.

10. UBAC2 was found to be phosphorylated by MARK2. In Fig. 5D phosphorylation of WT and S223A variants of UBAC2 were tested in control and MARK2 KD conditions. The experiment should be repeated including the phosphomimic S223D variant.

Response: We have examined the phosphorylation of UBAC2 S223D mutant by *MARK2* depletion, our results showed that the *MARK2* enhanced phosphorylation of UBAC2 was abolished when the S223 site was substituted with either alanine (A) or aspartate (D) (**New Figure 13**).

11. *The finding that UBAC2 expression in mice protects against inflammation in a model of DSS induced colitis are interesting, but the section appears to be disconnected from the mechanistic characterization of UBAC2. The authors should expand on the rationale for this experiment and analyze the UBAC2 mutants that are associated with human diseases using the RFP-GFP-KDEL construct.*

Response: We thank the reviewer for his interest in UBAC2 in protecting against colitis. The dysregulation of ER-phagy leads to the accumulation of unfolded or misfolded proteins in the ER lumen, known as ER stress, which activates the unfolded protein response (UPR) (Wang & Kaufman, 2016). The pathways activated by the ER stress response induce sterile inflammation (Garg *et al.*, 2012). To study the physiologic role of UBAC2-directed ER-phagy, we treated mice with dextran sulfate sodium (DSS) to instigate colonic inflammation. The roles of somatic mutations near the LIR region of UBAC2 were firstly examined. Our results showed that UBAC2 bearing R277C, F279S, or G293S mutant decreased the association with GABARAP (**New Figure 14A, also see Fig. 7A**). We then checked the roles of these UBAC2 mutants in ER-phagy flux, finding that the enhanced cleavage of RFP fragment by UBAC2 was abrogated by UBAC2 bearing these somatic mutations (**New Figures 14B and 14C, also see Fig. 7B,C**). We employed AAV-mediated genetic approaches combined with acute ulcerative colitis (UC) model to determine the role of UBAC2-mediated ER-phagy, and found that UBAC2 bearing somatic mutations near the LIR region increased the ER stress response and sterile inflammation. Our overall data suggest that the UBAC2-directed ER-phagy is essential for the maintain the ER homeostasis.

12. On page 9 line 13 it should read ... when S223 was mutated to alanine.

Response: We have modified our description as suggested by the reviewer.

13. Amount of puncta should be replaced by number of puncta in the text.

Response: We have corrected this part according to the reviewer's suggestion.

14. Why is no input (WCL) shown for the IgG control in Fig. 5B?

Response: For the Co-IP figures, the sample in the first lane and the second lane were used from the same one. The difference between lane 1 and lane 2 is that the samples in lane 1 used IgG, while the samples in lane 2 used anti-MARK2 antibody for immunoprecipitation. In the WCL, we didn't load the same sample in two lanes and have shown all our uncropped images in the Source Data Figures.

Response to the comments of Reviewer #3

1. The endoplasmic reticulum (ER), as the largest continuous membranous system within cells, plays a pivotal role in essential cellular processes such as protein synthesis, lipid metabolism, immune response, and organelle biogenesis. ER-phagy, the autophagy of the endoplasmic reticulum, is essential for maintaining ER homeostasis, and various ER-phagy receptors have been identified in previous studies. A significant scientific question in this field is whether there are additional ER-phagy receptors that respond to different ER stresses. In this manuscript, He and colleagues propose a novel ER-phagy receptor, UBAC2. Under inflammatory stimuli, UBAC2 undergoes phosphorylation modifications, leading to dimerization and activation of ER-phagy. In a dextran sodium sulfate (DSS)-induced inflammatory model, this biological process alleviates inflammation. Overall, the data in this manuscript are of high quality, the figures are pleasing, and the experimental phenotypes are evident. However, significant flaws exist in some experimental designs and logical coherence, making it potentially unsuitable for publication in the Embo Journal.

Response: We appreciate the reviewer for the comment that "Overall, the data in this manuscript are of high quality, the figures are pleasing, and the experimental phenotypes are evident". We have performed a series of experiments suggested by the reviewer and we believe that our revised manuscript is largely improved and can meet the standard for publication in the EMBO Journal.

2. *CCPG1* was previously considered one of the ER-phagy receptors in earlier studies. Authors observed a significant increase in its protein levels under conditions of TG treatment, using this as a crucial indicator in their screening work. What is the underlying mechanism for this? Authors maybe should present all previously recognized ER-phagy receptors and other clearly evident ER-phagy substrates in this context.

Response: CCPG1 has been identified as a binding partner of GABARAP to protect against ER luminal protein aggregation and consequent unfolded protein response hyperactivation through ER-phagy. We found that the transcription of *CCPG1*, but not *UBAC2* was up-regulated upon TG treatment (**New Figures 15A and 15B**). Our results were consistent with previous study which reported that *CCPG1* gene is inducible by the unfolded protein response (Smith *et al*, 2018). The up-regulated transcription of *CCPG1* gene caused the increased accumulation of CCPG1 in its protein level (**New Figure 15C**). It was reasonable that we showed CCPG1 as a positive control in our screening work. We also found that several well-known ER markers or ER-phagy substrates, such as RTN4, CANX, TEX264, PTPN1, and UFSP2 through mass spectrometric analysis. Since long-time biotin treatment inhibits autophagy and elicits ER stress to differentially regulate adipocyte lipid and protein synthesis (Selvam *et al*, 2019), we have labeled the proteins in ER membrane (ERM)-facing cytosol using PhastID with biotin treatment for only 1 hr. Although multiple ER-phagy receptors exist in mammal cells, their expression patterns, tissue distributions and involved signaling pathways are different. It can be hard to identify all the reported ER-phagy receptors with biotin labeling for 1 hr in HeLa cells.

3. Authors demonstrates the co-localization of UBAC2 with proteins such as ATG16L, LAMP1, GABARAP, etc., and suggests that the starvation treatment results in an increased co-localization, possibly due to a significant increase in the abundance of the associated proteins.

Response: To resolve the reviewer's concern, we have presented better images for our confocal microscopic analysis. To compare the co-localization between UBAC2 and components of autophagy machinery, we have added Baf A1 to inhibit the autolysosomal degradation of UBAC2 in these experimental settings. Baf A1 treatment resulted in the equality of protein abundances of UBAC2 and components of autophagy machinery, thus makes the samples suitable for further confocal analysis. Our updated data indicated that the co-localization between UBAC2 and components of autophagy machinery was significantly enhanced with autophagy activation. Also see Response to Comment #2 of Reviewer #1.

4. As a calcium ion inhibitor mediated by SERCAs, what is the direct mechanism of TG on UBAC2? Does UBAC2 directly bind to TG molecules, or does it depend on the functionality of other ER-phagy receptors?

Response: We used biotin-labeled TG and found that unlike sarcoplasmic reticulum Ca²⁺ ATPase 2 (SERCA2) (Paula & Ball, 2004; Zhong & Inesi, 1998), UBAC2 could not directly interact with TG (**New Figure 16A**). To study whether the role of UBAC2 relies on other known ER-phagy receptors, Co-IP and immunoblot analysis revealed that UBAC2 had only a weak interaction with FAM134B. However, the UBAC2-FAM134B association was not altered by starvation-induced autophagy activation (**New Figure 6A**). Confocal analysis indicated that FAM134B could partially colocalize with UBAC2 (**New Figure 6B**). We knocked down the expression of *FAM134B* and observed that UBAC2-directed ER-phagy was not affected by *FAM134B* deficiency. Our results suggest that the function of UBAC2 in ER-phagy flux receptor does not depend on FAM134B, ATL3, SEC62, RTN3, CCPG1, and TEX264. Also see Response to Comment #2 of Reviewer #2. We found that TG treatment activated the phosphorylation of MARK2 (**New Figure 16B**), and we think TG treatment may provide the activation signal upstream MAKR2-UBAC2 axis.

5. Throughout the entire manuscript, authors lack a comparison of UBAC2 with all currently identified ER-phagy receptors. Does UBAC2 function independently of other ER-phagy receptors? What is the proportion of UBAC2's involvement in the entire ER-phagy process?

Response: We found that UBAC2 had only a weak interaction with FAM134B, but not the other known ER-phagy receptors (including FAM134B, ATL3, SEC62, RTN3, CCPG1 and TEX264) (New Figure 6A, also see Fig. 3K). UBAC2-directed ER-phagy was not altered in *FAM134B*-deficient cells (New Figure 6C, also see Fig. 3L), suggesting that the function of UBAC2 in ER-phagy is relatively independent. To further reveal the proportion of UBAC2 participates in entire ER-phagy, we further determined the relative importance of these receptors by siRNA-mediated knockdown of six out of the seven receptors. Simultaneous depletion of *TEX264*, *FAM134B*, *CCPG1*, *RTN3*, *SEC62*, and *ATL3* (not *UBAC2*) caused a partial reduction in the cleavage of the ER-phagy reporter, but a significant level (>50%) of ER-phagy activity remained in THP-1 Mφs (New Figures 17A and 17B, lane 2). Cells expressing one of the other ER-phagy receptors (all lacking UBAC2) showed much lower ER-phagy activity (New Figure 17A and 17B, lanes 3–8). The data showed that deletion of *UBAC2* causes the largest inhibition of ER-phagy in THP-1 Mφs, indicating that although there is some redundancy among the known ER-phagy receptors, UBAC2 is an essential ER-phagy receptor. It is notable that a dozen of ER-phagy receptors exist in mammals, and these receptors may be redundant in some cases but have at least partially distinct functions under different circumstances. We found that the activation of MARK2 signaling can promote UBAC2-directed ER-phagy, and MARK2-UBAC2 axis is essential to restrain the inflammatory response. Our results uncover a newly identified ER-phagy process generated by MARK2-UBAC2 signaling.

6. Authors attempt to elucidate in the manuscript that the dimerization of UBAC2 is a functional form for its activity. What significance does this dimerization hold, and in essence, why does UBAC2 require dimerization to function? Is dimerization dependent on its transmembrane domain? Can the analysis of dimerization forms provide additional truncated data? Can it offer *in vitro* data or data from Native gel experiments?

Response: To resolve the reviewer's concern on the dimerization of UBAC2. We first generated the truncated form of UBAC2, the N terminal (NT, 1–91 aa; localizes in the ER lumen), the transmembrane domain (TMD, 92–184 aa), and the cytosolic C terminal (CT, 185–344 aa) (**New Figure 18A**). We first studied the interaction between the UBAC2 domain and full-length UBAC2, finding that the NT was not necessary for the interaction. Both the TMD and the cytosolic CT could associate with the full-length form of UBAC2. However, the interaction between the TMD of UBAC2 and full-length UBAC2 was not affected by TG treatment (**New Figure 18B**). We then investigated the assembly of purified UBAC2 into oligomerization through chemical cross-linking using disuccinimidyl suberate (DSS) and consequent western blot showed that both the TMD and the cytosolic CT could form dimers *in vitro* (**New Figure 18C**). The dimerization of the transmembrane truncate UBAC2 was not affected by TG treatment (**New Figure 18D**). Collectively, these results suggest that while the TMD can facilitate UBAC2 dimerization, the formation of UBAC2 dimers in response to thapsigargin-induced ER stress relies on the C-terminal (CT) region of UBAC2. To further deduce the significance of UBAC2 dimerization holds, we employed AlphaFold Protein Structure Database and High Ambiguity Driven protein-protein DOCKing (HADDOCK2.4) website to study the interaction between UBAC2 and GABARAP. We found that an interactive interface between GABARAP and the cytoplasmic CT of UBAC2 (**New Figure 18E**). The cytosolic tails of dimerized UBAC2 tend to “lock” GABARAP, consequently, dimerized UBAC2 can be better captured by the GABARAP-associated autophagosomes.

7. The phosphorylation sites identified by authors are distant from the ER membrane. How does phosphorylation by MARK2 affect the dimerization of UBAC2? What is the upstream signal of this process? Does the activation of MARK2 lead to an increase in ER-phagy levels?

Response: From HADDOCK2.4 website predicted result of dimerized UBAC2-GABARAP complex, we found the cytosolic tails of dimerized UBAC2 tend to “lock” GABARAP, consequently, dimerized UBAC2 can be better captured by the GABARAP-associated autophagosome (**New Figure 18E**). The phosphorylated S223 located at the variable region of UBAC2 makes the cytosolic tail of UBAC2 more flexible and easier bind to the CT of homo-UBAC2. The phosphorylation of many proteins, such as the key immune factors IRF3 and STING, can promote the dimerization and tetramerization of IRF3 and STING, respectively (Dalskov *et al*, 2020). We detected the activation of MARK2 signaling and found that that PEP005, a PKC δ activator treatment could promote the activation of PKC δ -MARK2 signaling axis and ER-phagy flux. However, the PEP005 induced ER-phagy flux was significantly abolished in UBAC2-deficient cells (**New Figure 19, also see Fig. EV4M**). We think the activation of MARK2 is an upstream signaling of UBAC2-directed ER-phagy.

8. In Figure 6, it would be more suitable for authors to use Tunicamycin rather than Thapsigargin to validate the proposed model.

Response: We have conducted a series of new experiments to detect the roles of UBAC2 in the UPR and inflammatory responses under tunicamycin (TM) treatment. Our results revealed that WT UBAC2, but not UBAC2 LIRm suppressed the transcription of ER-stress induced genes under TM treatment (**New Figure 20A, also see Fig. 6C**). Moreover, *UBAC2* KO enhanced the expression of ER-stress induced transcripts (**New Figure 20B, also see Fig. 6D**). The transcription of inflammatory cytokines induced by TM treatment could be suppressed by overexpression of WT UBAC2, but not UBAC2 LIRm (**New Figure 20C, also see Fig. 6E**). *UBAC2* KO increased the mRNA levels of inflammatory cytokines induced by TM treatment (**New Figure 20D, also see Fig. 6F**). Consistently, the suppressed production of IL6, TNF- α , and IL-1 β was abrogated when the LIR of UBAC2 was mutated (**New Figure 20E, also see Fig. 6G**). Moreover, ELISA results showed that the release of IL6, TNF- α , and IL-1 β was increased in *UBAC2* KO cells (**New Figure 20F, also see Fig. 6H**). Our western blotting analysis also revealed that *UBAC2* KO increased the protein abundances of ER stress markers upon TM treatment and *UBAC2* depletion enhanced the phosphorylation of p65 induced by TM (**New Figure 5A, also see Appendix Fig S2G**). Additionally, only WT UBAC2, but not LIR mutated UBAC2 reduced the protein abundances of ER stress markers and the phosphorylation of p65 (**New Figure 5B, also see Appendix Fig S2G**). We then knocked down the expression of *MARK2* in UBAC2-inducible cells, observing that the suppression of ER-stress induced transcripts by UBAC2 was abrogated by *MARK2* deficiency (**New Figure 20G, also see Fig. 6I**). Consistently, the inhibitory transcription of inflammatory genes by UBAC2 was also abolished in *MARK2*-deficient cells (**New Figure 20J**). Altogether, these data indicate that UBAC2 serves as ER-phagy receptor to suppress UPR and ER stress-associated inflammatory responses.

9. In Figure 7, in the colitis model experiments, it appears that authors need to supplement the differences between UBAC2 KO mice and WT mice, and to perform rescue experiments with various mutants in UBAC2 KO. Because UBAC2 exists in a dimeric form, and the authors did not resolve the details of this form, the endogenous UBAC2 in WT mice and the expression of various mutant forms may introduce interference.

Response: At present, we have no *Ubac2* KO mice, it will take more than one year to generate enough amounts of KO mice for further study. For time limitation, the editor agreed with us to perform the *in vivo* study using AAV-mediated knockdown system. Briefly, we overexpressed human UBAC2 mutants in AAV-delivered shRNA *Ubac2* knockdown mice to largely avoid the interference caused by endogenous mouse UBAC2 (New Figure 21A, also see Fig. 7D). We transduced human UBAC2 mutants in mice and found DSS mice expressing mutated UBAC2 experienced apparent body weight loss compared with mice expressing WT UBAC2 (New Figure 21B, also see Fig. EV5H). Consistently, the UBAC2 mutated DSS mice exhibited a remarkable increase of the shortening of colon lengths, and displayed much worse consistency and morphology of colons with infiltrating inflammatory cells (New Figures 21C–F, also see Fig. 7E–I). qPCR analysis of the intestines revealed that overexpression of the mutated form of UBAC2 increased the transcription of ER-stress inducible genes as well as inflammatory genes in DSS mice (New Figure 21G, also see Fig. 7J). Collectively, our data demonstrate that UBAC2-mediated ER-phagy contributes to cellular homeostasis, thus preventing the occurrence and development of inflammatory diseases.

10. In the confocal experiments, the imaging of UBAC is blurry, and HeLa cell morphology is poor.

Response: For the confocal experiments, we have analyzed the cells without immunostaining or GFP/RFP transfection as negative controls, and adjusted the laser intensity always on the untreated samples to avoid overexposure. If we increase the fluorescence intensity for each experiment group, the edges of the HeLa cells will be sharper (New Figure 22). We have provided better confocal images and re-analyzed the differences.

11. *Figure 2L, the confocal image is misplaced.*

Response: We are sorry for this mistake and have corrected it.

12. *For all in vitro experiments, please supplement the target protein's purified samples with Coomassie Brilliant Blue staining or other data confirming protein purity.*

Response: We have added the Coomassie Blue staining (CBS) of our purified protein samples to confirm their purity as suggested by the reviewer. Our results showed the purity of protein was sufficient for Co-IP assay (**New Figure 23, also see Fig. 2J,N**).

13. *For the WB experiments under TG treatment, please include ER-phagy substrates as positive controls.*

Response: We have used REEP5 as substrate of ER-phagy according to the reviewer's suggestion (**New Figure 24, also see Fig. 1D,F**).

14. *Authors should statistically present the RFP-GFP confocal experiment as a ratio rather than simply counting the number of RFP points.*

Response: We have analyzed the ratio of the RFP-GFP confocal experiment as suggested by the reviewer (**New Figures 9A–C and 11C–H**).

15. *Figure 4G and I, under the same treatment conditions, there is a significant disparity in the number of UBCA2-positive puncta in the Mock group. This suggests experimental instability.*

Response: In fact, the experimental conditions in Figure 4G and 4I were different. There was no Baf A1 treatment in the Mock group in Figure 4G, while Baf A1 was added to all groups in Figure 4I.

16. *To substantiate MARK2 as the kinase for UBAC2, additional data such as in vitro phosphorylation need to be provided.*

Response: We performed *in vitro* phosphorylation assay, and found that incubating MARK2 with UBAC2 increased the phosphorylation of WT UBAC2, but not the S223A mutated form of UBAC2 *in vitro* (**New Figure 4B**). Also see Response to Comment #5 of Reviewer #1.

17. *Figure 5k, how can ER-tracker be used to quantify the size of the endoplasmic reticulum (ER)? Do the authors propose alternative and potentially more effective methods for characterizing ER morphology?*

Response: We have detected the expression of ER markers (including REEP5 and CLIMP-63) using immunoblotting (**New Figures 3B, 3C and 4C, also see Figs. 3F,I and 4L**). We believe the protein abundances of ER markers can represent the ER content. Also see Response to Comment #4 of Reviewer #1.

18. *Given authors' attempt to establish UBAC2 as a novel ER-phagy receptor and their belief that the punctate UBAC2 structures observed in co-localization experiments indicate autophagic functionality, it is necessary for the authors to supplement staining with LAMP1 and LC3 in all GFP-UBAC2 punctate structure experiments.*

Response: We have showed confocal images of GFP-UBAC2 puncta together with LAMP1 and GABARAP as the reviewer's suggestion (**New Figure 25A**). And we also analyzed the colocalization between Flag-tagged UBAC2 mutants and LAMP1 as well as GABARAP (**New Figure 25B**). Our results showed that the co-localization between UBAC2 and LAMP1/GABARAP was largely decreased when the LIR of UBAC2 was mutated. Moreover, the phosphorylation of UBAC2 at S223 was crucial for the puncta formation of UBAC2, as well as its co-localization with LAMP1/GABARAP.

19. *Some portions of DSS were incorrectly labeled as SDS.*

Response: We apologize for this carelessness and have corrected this mistake.

References

Chen YH, Huang TY, Lin YT, Lin SY, Li WH, Hsiao HJ, Yan RL, Tang HW, Shen ZQ,

Chen GC *et al* (2021) VPS34 K29/K48 branched ubiquitination governed by UBE3C and TRABID regulates autophagy, proteostasis and liver metabolism. *Nat Commun* 12: 1-19

Chino H, Hatta T, Natsume T, Mizushima N (2019) Intrinsically Disordered Protein TEX264 Mediates ER-phagy. *Molecular Cell* 74: 909-921

Dalskov L, Narita R, Andersen LL, Jensen N, Assil S, Kristensen KH, Mikkelsen JG, Fujita T, Mogensen TH, Paludan SR *et al* (2020) Characterization of distinct molecular interactions responsible for IRF3 and IRF7 phosphorylation and subsequent dimerization. *Nucleic Acids Res* 48: 11421-11433

Garg AD, Kaczmarek A, Krysko O, Vandenabeele P, Krysko DV, Agostinis P (2012) ER stress-induced inflammation: does it aid or impede disease progression? *Trends Mol Med* 18: 589-598

Ishii S, Chino H, Ode KL, Kurikawa Y, Ueda HR, Matsuura A, Mizushima N, Itakura E (2023) CCPG1 recognizes endoplasmic reticulum luminal proteins for selective ER-phagy. *Mol Biol Cell* 34: 1-18

Li W, Long Q, Wu H, Zhou Y, Duan L, Yuan H, Ding Y, Huang Y, Wu Y, Huang J *et al* (2022) Nuclear localization of mitochondrial TCA cycle enzymes modulates pluripotency via histone acetylation. *Nat Commun* 13: 1-15

Newman LE, Weiser Novak S, Rojas GR, Tadepalle N, Schiavon CR, Grotjahn DA, Towers CG, Tremblay M, Donnelly MP, Ghosh S *et al* (2024) Mitochondrial DNA replication stress triggers a pro-inflammatory endosomal pathway of nucleoid disposal. *Nat Cell Biol* 26: 194-206

Paula S, Ball WJ, Jr. (2004) Molecular determinants of thapsigargin binding by SERCA Ca²⁺-ATPase: a computational docking study. *Proteins* 56: 595-606

Selvam S, Ramaian Santhaseela A, Ganesan D, Rajasekaran S, Jayavelu T (2019) Autophagy inhibition by biotin elicits endoplasmic reticulum stress to differentially regulate adipocyte lipid and protein synthesis. *Cell Stress Chaperones* 24: 343-350

Smith MD, Harley ME, Kemp AJ, Wills J, Lee M, Arends M, von Kriegsheim A, Behrends C, Wilkinson S (2018) CCPG1 is a non-canonical autophagy cargo receptor essential for ER-Phagy and pancreatic ER proteostasis. *Dev Cell* 44: 217-232

Wang M, Kaufman RJ (2016) Protein misfolding in the endoplasmic reticulum as a conduit to human disease. *Nature* 529: 326-335

Wang Y, Subramanian M, Yurdagul A, Jr., Barbosa-Lorenzi VC, Cai B, de Juan-Sanz J, Ryan TA, Nomura M, Maxfield FR, Tabas I (2017) Mitochondrial fission promotes the continued clearance of apoptotic cells by macrophages. *Cell* 171: 331-345

Zhong L, Inesi G (1998) Role of the S3 stalk segment in the thapsigargin concentration dependence of sarco-endoplasmic reticulum Ca²⁺ ATPase inhibition. *J Biol Chem* 273: 12994-12998

Dear Dr. Jin,

Thank you for submitting your revised manuscript for consideration by the EMBO Journal. It has now been seen by two referees whose comments are enclosed. As you will see, whilst referee #2 supports, referee #3 provides a list of additional concerns that may be considered in the discussion section. Pending these satisfactory minor revisions, and the inclusion of standard additional antibody controls you may already have performed, we can proceed towards publication. There are some further minor editorial points as follows:

- please reorganize Source Data files to one file/folder per figure and ZIP for each main figure. For EV and/or appendix figures, ZIP together all source data,
- please provide p-values in the legends of figures 1e, k, m, o; 2e, i, m; 3c, e, h; 5i; 6c-j; 7c, e, g, i-j; EV 1a, f-g, i; EV 2b; EV 3f; EV 4j, l; EV 5b, d, f-g,
- please define scale bars for figures 3d, g, j; 4g, i, l, o; and 5h,
- please define black arrowheads in the legend of figures 3j; 4o

We include a synopsis of the paper (see <http://emboj.embopress.org/>). Please provide me with a general summary image, a two-sentence summary statement and 3-5 bullet points that capture the key findings of the paper.

Thank you for the opportunity to consider your work for publication. I look forward to your revision.

Yours sincerely,

William Teale

William Teale, PhD
Editor
The EMBO Journal
w.teale@embojournal.org

We realize that it is difficult to revise to a specific deadline. In the interest of protecting the conceptual advance provided by the work, we recommend a revision within 3 months (5th Oct 2024). Please discuss the revision progress ahead of this time with the editor if you require more time to complete the revisions. Use the link below to submit your revision:

Referee #2:

The authors have addressed all concerns and I recommend publication of this highly relevant and exciting study in the EMBO journal.

Referee #3:

Despite the author's diligent efforts in addressing the reviewers' questions and providing responses, some experimental results remain difficult to support the author's conclusions; the authors should include additional controls to increase the readers' confidence in the commercial antibodies used. Without this, I still believe that this manuscript may not yet be suitable for publication in the EMBO Journal.

1: What is the mechanism by which TG induces the accumulation of misfolded proteins? If the author suggests that the upregulation of CCPG1 is due to the accumulation of misfolded proteins, it would be beneficial to supplement the manuscript with data on changes in UPR pathway-related proteins.

2: There appears to be a significant difference in the number of punctate structures of LAMP1 before and after EBSS treatment. Clarification on this point would be helpful.

3: The explanation provided for the regulation of UBAC2 function by TG is not fully convincing. There seems to be no previous report of TG regulating MARK2 function. This issue might require further exploration.

4: The manuscript states that TG affects UBAC2 function by regulating MARK2 phosphorylation. However, it is unclear how TG-induced calcium channel dysfunction leading to ER stress activates MARK2. The CT domain, which loses its TMD structure, would dissociate from the ER membrane, so further explanation is needed on how the CT domain responds to TG-induced ER stress. AlphaFold Protein Structure cannot conclusively prove the author's claim that UBAC2 needs dimerization to bind to GABARAP. Therefore, this section of the data might not fully address the reviewers' questions.

5: Clarification on why the DSS treatment was changed from 5% to 3% is necessary. Additionally, an explanation of how the other known functions of UBAC2 in vivo are coordinated with its role as an ER-phagy receptor would be valuable.

6: The confocal images do not clearly illustrate the conclusions emphasized by the author. Perhaps conducting a GFP-UBAC2 cleavage experiment would provide more definitive evidence.

7: Including the original gel bands in the Coomassie Blue data would strengthen the manuscript.

8: Since the author believes that TG treatment induces ER-phagy by activating MARK2, providing additional positive controls would be helpful.

9: Some experimental procedures and materials could benefit from further detail to ensure completeness.

Dear Dr. Teale,

We would like to thank you and the reviewers for the positive and thoughtful comments and suggestions regarding our manuscript. A point-by-point response to the Reviewer #3' concerns is included below (see Appendix-I).

Particularly, we have revised our manuscript and supplemental material based on your perspicuous suggestions, including: (1) We have reorganized the Source Data files. (2) We have previously provided P values on each figure, so we have added “The statistical significance of the difference was analyzed by unpaired two-tailed Student's t test, and the P values were shown” in the figure legend parts. (3) We have added the descriptions of scale bars and arrowheads in the legends.

We have highlighted our modification with yellow in our text. We hope that our revised manuscript can meet the standard for publication in '*EMBO Journal*', and we look forward to hearing from you.

Sincerely,

Shouheng Jin, Ph.D.

Associate Professor of Department of Biochemistry

School of Life Sciences

Sun Yat-sen University

(The corresponding author)

Appendix-I below

Response to the comments of Reviewer #3

Comment 1: Despite the author's diligent efforts in addressing the reviewers' questions and providing responses, some experimental results remain difficult to support the author's conclusions; the authors should include additional controls to increase the readers' confidence in the commercial antibodies used. Without this, I still believe that this manuscript may not yet be suitable for publication in the EMBO Journal.

Response: The antibodies we used have all been validated by each manufacturer and further been validated in our previous experiments using WB/IF/IP.

Comment 2: What is the mechanism by which TG induces the accumulation of misfolded proteins? If the author suggests that the upregulation of CCPG1 is due to the accumulation of misfolded proteins, it would be beneficial to supplement the manuscript with data on changes in UPR pathway-related proteins.

Response: Thapsigargin (TG) acts to deliver a stress response by inhibiting Ca^{2+} uptake by the ER (Richter *et al*, 2016), thus causing the ER stress. The mechanism by which TG induces the accumulation of misfolded proteins may be an interesting question, but it is not the issue that the UBAC2 story touched. We have not stated that the upregulation of CCPG1 is due to the accumulation of misfolded proteins. We put CCPG1 as an example since it serves as a famous ER resided protein and its expression can be upregulated by TG-induced ER stress.

Comment 3: There appears to be a significant difference in the number of punctate structures of LAMP1 before and after EBSS treatment. Clarification on this point would be helpful.

Response: It has been reported that the termination of autophagy and reformation of lysosomes are regulated by mTOR. Nutrient deprivation of mammalian cells, multiple lysosomes fuse with each autophagosome. After 4 hr, essentially all lysosomes were consumed into fewer and larger autolysosomes that expressed lysosomal-associated membrane protein 1 (LAMP1) (Yu *et al*, 2010). Our results were consistent with

previous study.

Comment 4: The explanation provided for the regulation of UBAC2 function by TG is not fully convincing. There seems to be no previous report of TG regulating MARK2 function. This issue might require further exploration.

Response: As mentioned by the reviewer, there is no report about the regulation of TG on MARK2. This can be a fascinating issue that need to be further studied, we have added more discussion on the regulation of MARK2 during ER stress in our manuscript on page 15.

Comment 5: The manuscript states that TG affects UBAC2 function by regulating MARK2 phosphorylation. However, it is unclear how TG-induced calcium channel dysfunction leading to ER stress activates MARK2. The CT domain, which loses its TMD structure, would dissociate from the ER membrane, so further explanation is needed on how the CT domain responds to TG-induced ER stress. AlphaFold Protein Structure cannot conclusively prove the author's claim that UBAC2 needs dimerization to bind to GABARAP. Therefore, this section of the data might not fully address the reviewers' questions.

Response: The regulation of TG on MARK2 can be an interesting project that need to be studied in the future, we have added more discussion on this in our manuscript on page 15.

Comment 6: Clarification on why the DSS treatment was changed from 5% to 3% is necessary. Additionally, an explanation of how the other known functions of UBAC2 in vivo are coordinated with its role as an ER-phagy receptor would be valuable.

Response: The mice with two steps of AAV-mediated transduction could be frailer, so we used 3% DSS to induce colitis. This low dosage of DSS was no as harmful as 5% DSS did. In fact, 3% to 5% DSS is reasonable for chronic ulcerative colitis mice model (Huang *et al*, 2014; Zhou *et al*, 2023).

Comment 7: The confocal images do not clearly illustrate the conclusions emphasized by the author. Perhaps conducting a GFP-UBAC2 cleavage experiment would provide more definitive evidence.

Response: We think this is a good suggestion and we will conduct such experiment in our study in the future.

Comment 8: Including the original gel bands in the Coomassie Blue data would strengthen the manuscript.

Response: We have provided gel bands in the Coomassie Blue for all our *in vitro* experiment to demonstrate the purity of our protein samples. The raw data of Coomassie Blue data have also been uploaded in our Source Data.

Comment 9: Since the author believes that TG treatment induces ER-phagy by activating MARK2, providing additional positive controls would be helpful.

Response: As mentioned by the reviewer, the study on the regulation of MARK2 in ER-induced ER-phagy is quite limited, so we have no positive controls can be used.

Comment 10: Some experimental procedures and materials could benefit from further detail to ensure completeness.

Response: We thank the reviewer for this suggestion. In the case of not affecting the length of the manuscript, we have supplemented the details of method.

References

Huang LY, He Q, Liang SJ, Su YX, Xiong LX, Wu QQ, Wu QY, Tao J, Wang JP, Tang YB *et al* (2014) ClC-3 chloride channel/antiporter defect contributes to inflammatory bowel disease in humans and mice. *Gut* 63: 1587-1595

Richter M, Vidovic N, Honrath B, Mahavadi P, Dodel R, Dolga AM, Culmsee C (2016) Activation of SK2 channels preserves ER Ca²⁺ homeostasis and protects against ER stress-induced cell death. *Cell Death Differ* 23: 814-827

Yu L, McPhee CK, Zheng L, Mardones GA, Rong Y, Peng J, Mi N, Zhao Y, Liu Z, Wan

F *et al* (2010) Termination of autophagy and reformation of lysosomes regulated by mTOR. *Nature* 465: 942-946

Zhou Y, Ji G, Yang X, Chen Z, Zhou L (2023) Behavioral abnormalities in C57BL/6 mice with chronic ulcerative colitis induced by DSS. *BMC Gastroenterol* 23: 1-10

Dear Dr. Jin,

I am pleased to inform you that your manuscript has been accepted for publication in the EMBO Journal.

Congratulations! I am very pleased to see this work in The EMBO Journal?

Yours sincerely,

William Teale

William Teale, PhD
Editor
The EMBO Journal
w.teale@embojournal.org
